# Fast geometric learning with symbolic matrices

**Jean Feydy**[*]
Imperial College London
jfeydy@ic.ac.uk

**Joan Alexis Glaunès**[*]
Université de Paris
alexis.glaunes@parisdescartes.fr

**Benjamin Charlier**[*]
Université de Montpellier
benjamin.charlier@umontpellier.fr

**Michael M. Bronstein**
Imperial College London / Twitter
m.bronstein@imperial.ac.uk

## Abstract

Geometric methods rely on tensors that can be encoded using a symbolic formula and data arrays, such as kernel and distance matrices. We present an extension for standard machine learning frameworks that provides comprehensive support for this abstraction on CPUs and GPUs: our toolbox combines a versatile, transparent user interface with fast runtimes and low memory usage. Unlike general purpose acceleration frameworks such as XLA, our library turns generic Python code into binaries whose performances are competitive with state-of-the-art geometric libraries – such as FAISS for nearest neighbor search – with the added benefit of flexibility. We perform an extensive evaluation on a broad class of problems: Gaussian modelling, K-nearest neighbors search, geometric deep learning, non-Euclidean embeddings and optimal transport theory. In practice, for geometric problems that involve $10^3$ to $10^6$ samples in dimension 1 to 100, our library speeds up baseline GPU implementations by up to two orders of magnitude.

## 1 Introduction

Fast numerical methods are the fuel of machine learning research. Over the last decade, the sustained development of the CUDA ecosystem has driven the progress in the field: though Python is the lingua franca of data science and machine learning, most frameworks rely on efficient C++ backends to leverage the computing power of GPUs [1, 86, 101]. Recent advances in computer vision or natural language processing attest to the fitness of modern libraries: they stem from the mix of power and flexibility that is provided by PyTorch, TensorFlow and general purpose accelerators such as XLA.

Nevertheless, important work remains to be done. Geometric computations present a clear gap in performances between Python and C++: notable examples are implementations of point cloud convolutions or of the nearest neighbor search [65, 76]. To scale up geometric computations to real-world data, a common practice is therefore to replace the compute-intensive parts of a Python code by handcrafted CUDA kernels [35, 60, 92]. These are expensive to develop and maintain, which leads to an unfortunate need to compromise between ease of development and scalability.

To address this issue, we present KeOps: an extension for PyTorch, NumPy, Matlab and R that combines the speed of a handcrafted CUDA kernel with the simplicity of a high level language. Our toolbox optimizes map-reduce operations on generalized point clouds and provides transparent support for distance-like matrices, as illustrated in Figure 1. The resulting computations are fully differentiable and have a negligible memory footprint. Their runtimes are competitive with state-of-the-art CUDA libraries when they exist, and peerless in the many use cases that are not covered by existing implementations. Our library fits seamlessly within existing codebases and provides a sizeable performance boost to a wide range of methods. Among other applications, we present optimal transport solvers and geometric operators in hyperbolic spaces which are orders of magnitude faster than the state-of-the-art. We believe that our library is an important addition to the existing arsenal of tools and will have a stimulating impact on machine learning research.

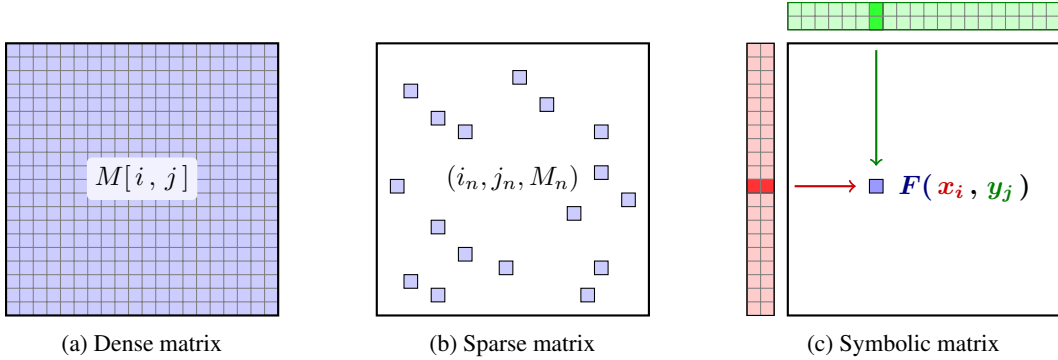

| (a) Dense matrix | (b) Sparse matrix | (c) Symbolic matrix |

Figure 1: Machine learning frameworks understand variables as matrices, also known as tensors. (a) These are usually **dense** and encoded as explicit numerical arrays $(M_{i,j}) = (M[i,j]) \in \mathbb{R}^{N \times M}$ that can have a large memory footprint. (b) Alternatively, some operators can be encoded as **sparse matrices**: we store in memory the indices $(i_n, j_n)$ and values $M_n = M_{i_n, j_n}$ that correspond to a small number of non-zero coefficients. Reduction operations are then implemented using indexing methods and scattered memory accesses. (c) We provide support for a third class of tensors: **symbolic matrices** whose coefficients are given by a formula $M_{i,j} = F(x_i, y_j)$ that is evaluated on data arrays $(x_i)$ and $(y_j)$. Reduction operations are implemented using parallel schemes that compute the coefficients $M_{i,j}$ on-the-fly. We take advantage of the structure of CUDA registers to bypass costly memory transfers and achieve optimal runtimes on a wide range of applications.

## 2   Related work

Machine learning and data science applications often encounter the problem of computing the proximity or distance between data samples. Given $x_1, \dots, x_N$ and $y_1, \dots, y_M \in \mathbb{R}^D$ two clouds of N and M points in dimension D, the bottleneck of many methods is an **interaction step** of the form:

$$a_i \; \leftarrow \; \overset{M}{\underset{j=1}{\square}} \, F(i, j, x_i, y_j) \,, \quad \forall i \in [\![1, N]\!] \,, \tag{1}$$

where $F$ is a vector-valued formula and $\square$ is an associative reduction operator, e.g. a sum or a minimum. This paper is part of a large body of work that lowers the $\mathcal{O}(NM)$ computational cost of such an operation: we now recall the main approaches to this problem.

**Sparse matrices.**    A first strategy is to prune out negligible terms: for every index $i$, we perform the reduction (1) on a subset of neighbors $\mathcal{N}(i) \subset [\![1, M]\!]$. As illustrated in Figure 1, this method is akin to using sparse matrices: the neighborhood structure is usually understood as a connectivity matrix that comes from a triangle mesh or a K-nearest neighbors (KNN) graph [16, 69, 114]. This method can be used whenever the operation $F$ is local but has a major limitation: at a low level, truncated reductions rely on random memory accesses that do not stream well on GPUs [25, 82]. Consequently, speed-ups are only achieved if the neighborhoods $\mathcal{N}(i)$ are orders of magnitude smaller than the full set of indices $[\![1, M]\!]$ – a condition that is often too restrictive and cannot be satisfied.

**Nearest neighbor finders.**    Going further, the implementation of KNN queries is itself a geometric problem in the mould of (1). When the datasets $(x_i)$ and $(y_j)$ have a small intrinsic dimension, efficient schemes can outperform brute-force approaches [11, 31, 49, 53, 65, 77, 85]. Unfortunately, these methods rely on pre-computations that are too expensive to be performed at every iteration of a training loop. Reference implementations also tend to lack flexibility and only support a handful of metrics: for instance, in spite of a strong interest for hyperbolic embeddings in the machine learning literature [72, 83], Poincaré metrics are not supported out-of-the-box by standard libraries.

**Approximated convolutions.**    When the reduction $\square$ is a sum and $F(i, j, x_i, y_j) = k(x_i - y_j) = K_{i,j}$ is a translation-invariant kernel, the interaction (1) is understood as a discrete convolution. To speed up this operation, a first idea is to rely on low-rank decompositions of the kernel matrix $(K_{i,j})$ [90, 111, 116]. Multiscale schemes can be used to handle singular kernels [7, 8, 50, 52, 115] or compress generic operators [5, 17, 55]. Alternatively, semi-Eulerian methods rely on intermediate

grid representations to leverage fast Fourier transforms or convolution routines [32, 51, 76]. These approaches can achieve dramatic speed-ups but tend to require a significant amount of tuning for each kernel $k$. They work best when the latter is smooth or is defined on a space of dimension $D \leqslant 3$.

**Acceleration frameworks.** In contrast to mathematical approaches, several compilation frameworks have been designed to speed-up machine learning architectures. Modern toolboxes accelerate a wide range of operations but are not geared towards geometric problems: most of them keep a focus on distributed learning [63, 64, 97, 108] or image processing and dense tensor manipulations [22, 58, 74, 105]. TVM [22] and CuPy [84] are the two libraries which are closer to our work: they both provide partial support for symbolic tensors. However, they have limited support for automatic differentiation and require the use of a custom low-level syntax to produce efficient binaries.

## 3 Motivation

**Requirements for geometric data analysis and learning.** None of the aforementioned methods are fully suited for modern research in geometric data analysis and machine learning. Let us briefly explain why. First of all, some acceleration schemes do not stream well on GPUs or have to rely on expensive pre-computations: hierarchical matrices [55] or advanced nearest neighbor finders [77] can hardly be used in the training loop of a neural network. Other strategies make strong assumptions on the properties of the convolution filter $k$ or on the dimension and geometry of the ambient feature space. These restrictions make existing tools cumbersome to use in deep learning, where one wishes to have modelling freedom, e.g. w.r.t. the choice of the embedding space geometry and dimension. Finally, most acceleration frameworks for Python expect users to be knowledgeable on GPU parallelism or do not support automatic differentiation. The bottomline is that most existing tools are not ready to be used by a majority of researchers in the community.

**A gap in the literature.** In order to tackle these issues, the developers of deep learning libraries have recently put an emphasis on just-in-time compilation for neural networks. For instance, the recent PyTorch JIT and XLA engines enable operator fusion and unlock performance speed-ups for research code [15, 86]. These *general purpose* compilers are fully transparent to users and show promise for a wide range of applications. Nevertheless, they fall short on geometric computations along the lines of (1). This is most apparent for nearest neighbor search [36, 60, 65], matrix-vector products with kernel matrices and message passing methods on point clouds [35, 36, 102], where one still has to develop and maintain custom CUDA kernels to achieve state-of-the-art performance.

**Symbolic matrices.** We notice that all the aforementioned methods rely on reductions of an N-by-M matrix $(M_{i,j}) = (F(i, j, x_i, y_j))$ that is often **too large to be stored in memory as a dense tensor**. Acknowledging the fact that memory management is a bottleneck for tensor programs, we choose to focus on the fundamental concept of symbolic matrices, illustrated in Figure 1. For the first time, we provide support for this abstraction on the GPU with all the desirable features of a deep learning library: a math-friendly interface, high performance, transparent support for batch processing and automatic differentiation. The example below is representative of our user interface:

```python
from torch import rand, autograd      # NumPy, R and Matlab are also supported
from pykeops.torch import LazyTensor  # Symbolic wrapper for PyTorch Tensors

# Setup data on the CPU and/or GPU with shapes (N,D), (M,D), (M,E):
N, M, D, E = 10 ** 5, 10 ** 6, 50, 100
x, y, b = rand(N, D, requires_grad=True), rand(M, D), rand(M, E)

# Perform arbitrary symbolic computations:
x_i = LazyTensor(x.view(N, 1, D))     # (N,D) Tensor -> (N,1,D) Symbolic Tensor
y_j = LazyTensor(y.view(1, M, D))     # (M,D) Tensor -> (1,M,D) Symbolic Tensor
D_ij = ((x_i - y_j) ** 2).sum(dim=2)  # (N,M) Symbolic matrix of squared distances.
K_ij = (- D_ij).exp()                 # (N,M) Symbolic Gaussian kernel matrix.

# Come back to genuine torch Tensors with reductions on dimensions 0 and 1:
nn = D_ij.argKmin(K=10, dim=1)  # K-NN search: (N,10) array of integer indices.
a  = K_ij @ b  # Kernel matrix-vector product: (N,M) * (M,E) = (N,E)
[g_x] = autograd.grad((a ** 2).sum(), [x])  # Seamless backpropagation.
```

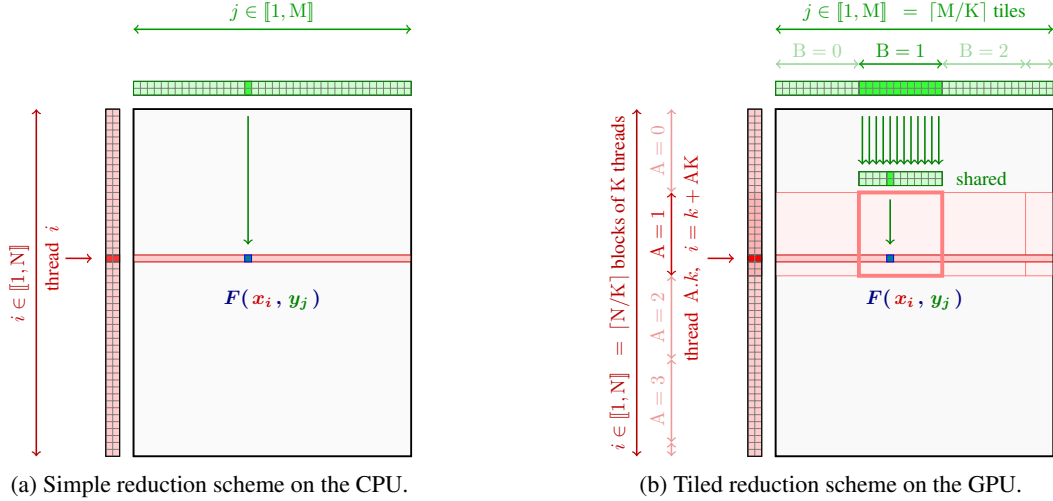

(a) Simple reduction scheme on the CPU.         (b) Tiled reduction scheme on the GPU.

Figure 2: We rely on fast parallel schemes to compute reductions of symbolic matrices, as in Eq. (1). (a) On the CPU, each thread $i$ computes a value $a_i$ by looping over the reduction index $j$ and consuming the values of $F$ **on-the-fly**. (b) On the GPU, we cut (a) in K-by-K tiles (where K is the CUDA block size) to leverage the low latency of the shared memory buffer and block-wise memory accesses. This ensures an optimal management of the $y_j$'s: we refer to our online documentation (`www.kernel-operations.io`) and to the N-body simulation chapter of [82] for details.

## 4   Implementation

**Creating a symbolic tensor.**   The entry point to our library is a `LazyTensor` wrapper that turns dense arrays into symbolic matrices (lines 9-10). We use standard operations to build up arbitrary formulas $F$: lines 11-12 update a symbolic representation of the matrices `D_ij` and `K_ij`. Our math engine is **lazy**, and defers the evaluation of formulae until reduction time. It is also **versatile**: we support batch dimensions, operator broadcasting and a wide range of elementary operations. Formulae can use an arbitrary number of variables $x_i^1, x_i^2, \dots$ and $y_j^1, y_j^2, \dots$ with varied dimensions.

**Parallel reduction schemes.**   Numerical computations take place at the reduction time (lines 15-16), when an associative operator $\square$ is called to perform an interaction step (1). To this end, we use a tiled map-reduce scheme that is detailed in Figure 2. Our C++ engine optimizes the use of registers to avoid costly memory transfers: the algorithm has $\mathcal{O}(\mathrm{NM})$ time complexity but does not allocate any buffer in the global device memory. For the sake of performance, we compile a specific binary for every new formula-reduction pair $(F, \square)$. These are stored in a cache directory for later use: the compilation is only done once. Unlike most acceleration frameworks, we do not expect the sizes of the data arrays $(x_i) \in \mathbb{R}^{\mathrm{N} \times \mathrm{D}}$ and $(y_j) \in \mathbb{R}^{\mathrm{M} \times \mathrm{D}}$ to be fully known at compile time: our binaries only rely on the feature dimensions D. This allows us to work with datasets of varying sizes, without having to re-compile binaries or sub-sample point clouds to an arbitrary point count. Notably, this allows our engine to process raw shape data on-the-fly.

**Pruning negligible interactions.**   Our library provides support for alternative reduction strategies that improve numerical accuracy [67] or increase GPU usage when $\mathrm{N} < \mathrm{M}$. Going further, our symbolic tensors support the specification of block-wise sparsity masks as optional attributes. Block reduction tiles are encoded using a collection of $(i_{\text{start}}, i_{\text{end}}, j_{\text{start}}, j_{\text{end}})$ tuples of indices that allow our engine to focus on a subset of the full collection of interaction pairs $[\![1, \mathrm{N}]\!] \times [\![1, \mathrm{M}]\!]$. They can be used to prune out negligible terms from symbolic reductions, without giving up on the contiguous memory accesses that make or break the performance of CUDA kernels.

Among high-level computing frameworks, this feature is unique to our library. Block-diagonal sparsity masks allow us to provide seamless support for batch processing, even in heterogeneous situations that are of interest for point cloud and mesh processing. Assuming that the data arrays have been pre-sorted in contiguous clusters, block-sparsity masks also enable the GPU implementation of fast multiscale methods such as the Barnes-Hut algorithm [7].

**Strengths and limitations.** At its heart, KeOps leverages the low Kolmogorov complexity of symbolic arrays: it can be used when the computational bottleneck of a method is an interaction step of the form (1). As we show next in the benchmarks of Section 5, KeOps is likely to offer gains on runtime and memory usage when the numbers of samples N and M range from $10^3$ to $10^7$.

The main limitation of KeOps stems from the overflow of CUDA registers in the reduction of Figure 2: these result in decreased performances on large feature vectors with $D > 100$. The problem is known as *register spilling*, with some documented work-arounds [25, 82]. Our priority for future developments is to improve performance for problems with $D \sim 1k$.

Another drawback is that we do not pre-ship binaries but instead rely on C++/CUDA compilers to run our kernels. To mitigate deployment issues and ease maintenance in the long run, we implement the core of the library in C++ and keep dependencies to a minimum [61]. In practice, compilation times range from 10 to 25 seconds when $D \leqslant 1,000$ and can be prohibitive for $D \geqslant 2,000$. Our library runs out-of-the-box on Linux or Mac configurations that provide a CUDA environment, e.g. fresh Google Colab sessions.

## 5 Experiments and Applications

**Configuration.** We now showcase our toolbox on a wide range of machine learning problems. All benchmarks were performed on a workstation equipped with 8 Intel Xeon Gold 6142 CPU @ 2.60GHz cores (16 threads), 128Gb of RAM and a Nvidia RTX 2080 Ti GPU with 11Gb of device memory. When relevant, we include comparisons with PyTorch JIT and JAX/XLA: just like our library, these two frameworks offer a transparent user interface and high performance on GPU. At the Python level, the implementations tested are rigorously equivalent to each other: we implement the same computations and only have to account for minor syntactic divergences. Whenever possible, we work with batches of samples and shapes to improve GPU usage.

**Notations, datasets.** In tables, "mem" stands for an out-of-memory error, "$\infty$" for a time that was too high to be recorded and "—" denotes a lack of available implementation. $L^1$ and $L^2$ denote the Manhattan and Euclidean metrics, respectively, $\langle\,,\rangle$ denotes the cosine similarity and $\mathbb{H}^D$ is the Poincaré metric on a hyperbolic space of dimension D [18, 21]. We perform numerical experiments with random normal samples and freely available datasets: digits from Scikit-Learn [87], Stanford dragon [26], ShapeNet [19]. MNIST [73], SIFT [62], GloVe-25 and GloVe-100 [88] were taken from the ANN-benchmarks repository [4], while HyperE-10 and HyperE-50 are hyperbolic embeddings processed from WordNet datasets [95].

### 5.1 Kernel Methods and Clustering

**Kernel methods.** Accelerating kernel methods is one of the core strengths of KeOps [20]. The code snippet of Section 3 shows how to perform a Gaussian convolution on point clouds and can be adapted to any other standard kernel. As detailed in the Supplementary Materials, symbolic `LazyTensors` can be transparently interfaced with the standard iterative solvers of the `scipy.sparse.linalg` library [66, 104]. This allows us to solve large-scale Gaussian regression problems ($N > 100k$-$1M$) in seconds on a single GPU [57, 78]. Going forward, KeOps provides solid numerical foundations for recent methods in the field that rely on kernel matrices [45, 80].

**Clustering.** As detailed in the Supplementary Materials, KeOps can be used to implement K-means clustering with several metrics. Going further, we use our symbolic `LazyTensors` to implement the standard Expectation Maximisation (EM) algorithm on a Gaussian Mixture Model (GMM) [56]: we estimate the weight, mean and covariance matrix of each of the K components. To this end, the EM algorithm strives to maximize the likelihood that is computed from a data sample of N points in $\mathbb{R}^D$. The most costly computations involve sum reductions of a K-by-N `LazyTensor` that encodes interactions between the K means of the clusters and the N sample points. Our results are summarized in Table 1. We show that KeOps performs well when computations involve variables of modest size: it can scale to large datasets in seconds, without memory overflows. This is most apparent when covariance matrices are diagonal and encoded as vectors of size D. In the second half of the table, we handle full D-by-D covariance matrices and notice a slow-down when $D \geqslant 10$.

## 5.2 Nearest neighbor search with any metric

**Syntax for a K-nearest neighbors query.** Our library enables the efficient implementation of brute-force KNN finders for any metric that is known in closed form. Users can apply the `argKmin(K=...)` reduction to extract the indices for the K-smallest values along the lines of any symbolic tensor:

```
1  # Turn (N,D) and (M,D) dense arrays into symbolic tensors:
2  x_i, y_j = LazyTensor(x.view(N,1,D)), LazyTensor(y.view(1,M,D))  # (N,1,D), (1,M,D)
3
4  # Compute the (N,M) symbolic matrix of distances:
5  E_ij = ((x_i - y_j) ** 2).sum(dim=2)     # Squared Euclidean metric
6  M_ij =  (x_i - y_j).abs().sum(dim=2)     # Manhattan distance
7  C_ij = 1 - (x_i | y_j)                   # Cosine similarity
8  H_ij = E_ij / (x_i[...,0] * y_j[...,0])  # Hyperbolic metric on the half-space
9
10 # Perform a K-NN search - in this case, for the hyperbolic metric:
11 indices = H_ij.argKmin(K=10, dim=1)  # Dense (N,K) array of integer indices
```

In the example above, the hyperbolic metric is defined on the half-space model of dimension D, with a positive first coordinate $x[0] > 0$ for feature vectors $x \in \mathbb{R}^D$ [18]. The mapping $x \mapsto \text{arcosh}(1 + x/2)$ is increasing, which allows us to work with the pseudo-distance $H(x_i, y_j) = \|x_i - y_j\|^2 / (x_i[0]y_j[0])$: similar acceleration tricks can be applied to e.g. the Poincaré ball metric.

**Exact and approximate nearest neighbor search.** The complexity of a KNN query depends on the number of points N and features D that are available for a given dataset: in the literature, two types of strategies have been designed to handle different scenarios. On the one hand, **exact** bruteforce schemes compute all pairwise distances between query and data points. These methods stream well on GPUs, have little to no parameters to tune and require no pre-processing of the data. On the other hand, **approximate** schemes leverage the structure of the dataset to prune out useless computations whenever possible. These methods are usually implemented on CPUs and tend to rely on hierarchical decompositions of the dataset that must be pre-computed ahead of the first KNN query. As discussed in Sections 2 and 3, approximate methods are tailored for large-scale retrieval whereas bruteforce schemes are generally more suited to geometric scenarios.

**Benchmarks.** To illustrate this behaviour, we compare our bruteforce reduction scheme to a selection of common baselines: bruteforce PyTorch and JAX implementations on the GPU; the popular FAISS library [65], with an approximate HNSW algorithm on the CPU [77] and two dedicated CUDA implementation on the GPU: an exact bruteforce scheme and the approximate IVF-Flat method. To showcase the variety of settings in which KNN queries are used throughout the literature, we run these methods on a collection of *random* normal samples and *structured* datasets. We use $N = 10^4$ to $10^7$ points with $D = 3$ to $784$ features: these lower and upper bounds let us represent the range of problems that are encountered in geometric data analysis, from shape processing to the manipulation of word embeddings. We stress that for larger datasets, approximate methods would clearly outperform our bruteforce approach.

Results are displayed in Table 2. A first observation is that on these medium-sized problems, our bruteforce GPU implementation is faster than approximate CPU methods to build a full KNN graph: even when FAISS-HNSW outperforms KeOps in terms of queries per second, this comes at the cost of a significant pre-processing time. A second remark is that our **generic** reduction engine for bruteforce computations is on par with the hand-crafted CUDA routines of the FAISS library: KeOps is less efficient than the FAISS-Bruteforce routine when $D \geqslant 50$ but is up to $\times 3$ faster in smaller dimensions. Crucially, we also provide the only competitive routines for non-Euclidean geometries, such as the increasingly popular hyperbolic metrics [83].

**Conclusion.** Overall, the performances of our C++ engine hold up to scrutiny: KeOps provides respectable runtimes on small and medium-sized problems (up to $N = 10^6$ and $D = 100$) with the added benefit of **flexibility**. These results attest to the effectiveness of our optimization techniques and bode well for the other computations that are supported by the KeOps engine. Going forward, we note that our library could probably be used as an efficient GPU backend for approximate KNN methods [31, 54, 118] and intend to provide the relevant tools for researchers in the field.

Table 1: Fitting a Gaussian Mixture Model: we perform 10 iterations of the standard EM algorithm with N points and K components in dimension D.

| Covariances | N | K | D | Sklearn | PyTorch | **Ours** |
|---|---|---|---|---|---|---|
| Diagonal | 50k | 100 | 5 | 2.9 s | **19.0 ms** | 33 ms |
| Diagonal | 500k | 1k | 5 | 286 s | 0.94 s | **0.22 s** |
| Diagonal | 5M | 10k | 5 | $\infty$ | mem | **11.38 s** |
| Diagonal | 500k | 1k | 50 | $\infty$ | mem | **2.96 s** |
| Diagonal | 5M | 10k | 50 | $\infty$ | mem | **245 s** |
| Full | 50k | 100 | 5 | 6.9 s | 0.23 s | **0.04 s** |
| Full | 500k | 1k | 5 | 830 s | mem | **0.362s** |
| Full | 50k | 100 | 20 | 16.0 s | mem | **0.84 s** |
| Full | 500k | 1k | 20 | $\infty$ | mem | **63 s** |

Table 2: KNN search: average queries per second with a dataset of N points in dimension D. We work with batches of 10k queries at a time and K = 10 neighbors. The first three columns correspond to schemes that are provided by the FAISS library; the approximate methods HNSW and IVF-Flat are tuned to provide a minimum recall of $90\%$ and optimize runtimes (we use the parameters of the ANN-benchmarks website as a first reference); when relevant, the pre-processing time is reported in parenthesis. We stress that choosing optimal parameters for the HNSW and IVF-Flat routines is a fairly complex problem: as non-specialists, we cannot guarantee that our experiments reflect the best level of performance that can be reached by these impressive methods. This is especially true for the IVF-PQ routine that is provided by FAISS and combines the IVF algorithm with a quantization method: it certainly fares even better than IVF-Flat on large-scale problems, but we found it to be very complex to use and opted to not include our unreliable benchmarks in this Table. High number of queries and low pre-processing is better: we highlight the column with the fastest time to process N queries and thus build a KNN graph for the dataset. (*) performed with smaller batch sizes when necessary to avoid memory overflows

| Dataset, metric | N | D | HNSW (CPU) | Bruteforce | IVF-Flat | PyTorch* | JAX* | **Ours** |
|---|---|---|---|---|---|---|---|---|
| Random, $L^2$ | 10k | 3 | 1.5e6 (0.15s) | 3.3e6 | 1.9e6 | 8.8e5 | 1.6e5 | **6.8e6** |
| Random, $L^2$ | 1M | 3 | 1.3e6 (25s) | 5.0e4 | **1.6e6** | 6.8e3 | 5.5e2 | 1.8e5 |
| Random, $L^2$ | 1M | 10 | 3.1e5 (53s) | 4.6e4 | **2.9e5** | 4.9e3 | 4.9e2 | 1.1e5 |
| Random, $L^2$ | 1M | 100 | 1.5e3 (540s) | **3.1e4** | mem | 9.0e2 | 4.0e2 | 1.4e4 |
| Random, $L^2$ | 10M | 100 | ($\infty$) | **3.2e3** | mem | $\infty$ | $\infty$ | 1.4e3 |
| Random, $L^1$ | 1M | 10 | — | — | — | 1.7e3 | 4.6e2 | **1.1e5** |
| Random, $L^1$ | 1M | 100 | — | — | — | 2.1e2 | 3.0e2 | **1.5e4** |
| MNIST, $L^2$ | 60k | 784 | 5.4e4 (14s) | 1.5e5 | **2.2e5** | 4.7e4 | 3.4e3 | 2.5e4 |
| MNIST, $L^1$ | 60k | 784 | — | — | — | 5.7e2 | 2.0e3 | **2.5e4** |
| GloVe-25, $\langle , \rangle$ | 1.2M | 25 | 7.7e4 (130s) | 4.2e4 | **1.6e5** | 2.7e3 | 4.1e2 | 4.8e4 |
| GloVe-100, $\langle , \rangle$ | 1.2M | 100 | 8.6e3 (480s) | 2.6e4 | **3.9e4** | 6.8e2 | 3.3e2 | 1.3e4 |
| Random, $\langle , \rangle$ | 1M | 10 | 1.8e5 (77s) | 4.6e4 | **2.7e5** | 5.2e3 | 5.3e2 | 1.5e5 |
| Random, $\mathbb{H}^D$ | 1M | 10 | — | — | — | 3.2e3 | 5.1e2 | **7.1e4** |

Table 3: Accelerating geometric deep learning architectures: shown is average training / inference time per shape on the ShapeNet dataset, with clouds of N = 2,048 points in dimension D = 3.

| | PointCNN $\rightarrow$ **Ours** | DGCNN $\rightarrow$ **Ours** |
|---|---|---|
| training | 254 ms $\rightarrow$ **128 ms** | 170 ms $\rightarrow$ **80 ms** |
| inference | 172 ms $\rightarrow$ **43 ms** | 109 ms $\rightarrow$ **20 ms** |

## 5.3 Geometric deep learning and geometric primitives

**Geometric deep learning.** Fast low-dimensional KNN search has important applications in the field of geometric deep learning [16], a popular class of algorithms now ubiquitous in 3D computer vision and graphics [110]. As a first illustration, we use KeOps to speed-up two popular networks for 3D point cloud segmentation: Point CNNs [75] and Dynamic Graph CNNs (DGCNN) [110]. We follow closely the original architectures for part segmentation on the ShapeNet dataset [19] and compare two different implementations: a reference PyTorch_Geometric code [36] and a hybrid PyTorch_Geometric+KeOps implementation, where KNN graphs are built as in Section 5.2. Results are summarized in Table 3: training denotes a full "forward + backward" pass through the network whereas inference is forward only, with a faster batch normalisation [59]. Note that the Deep Graph Library [109] relies on a bruteforce PyTorch implementation for KNN search and behaves like Py-Torch_Geometric on these problems. In practice, the switch to KeOps provides $\times 2$ and $\times 5$ speedups for training and inference times, respectively: the construction of KNN graphs becomes a negligible overhead, while the majority of the network runtime is spent on MLP and batch normalization layers.

**Geometric descriptors at all scales.** Table 4 shows the results of using KeOps to compute geometric features on generalized point clouds. In the interaction step (1), these correspond to the case where the reduction $\square$ is a (possibly weighted) average on a neighborhood of $x_i$ and the formula $F$ is a function of the difference $x_i - y_j \in \mathbb{R}^D$: the identity $x \mapsto x$, an outer product $x \mapsto xx^\top$ or a multi-layer perceptron with H hidden neurons and O output features. Neighborhoods are either defined through a 40-nearest neighbors query or by weighting each pair of points with a window of radius $\sigma$ such as:

$$k(x_i, y_j) = \exp(-\|x_i - y_j\|^2 / 2\sigma^2). \tag{2}$$

Our library is well suited for these computations and consistently outperforms the PyTorch and JAX baselines by up to two orders of magnitude. We remark that on clouds of N = 2,048 points, our bruteforce scheme (set radius) is up to an order of magnitude faster than the best sparse methods, which rely on scattered memory accesses to build neighborhoods as (N, K, D) arrays. A similar behavior is observed for chamfer and Energy distance computations [14, 99]. These results highlight the dichotomy between *contiguous* ("bruteforce") and *scattered* ("sparse") memory accesses, which are optimized separately by GPUs and are best suited to different types of computations. We stress that KeOps supports the specification of block-wise sparsity schemes, which let users implement tree-based pruning strategies to best leverage the structure of their problems.

**Conclusion.** Overall, our library enables the quick development of differentiable layers for geometric deep learning which are an order of magnitude faster than Python baselines. It factors out C++ boilerplate code for point cloud processing and lets researchers focus on their *models*. We thus believe that KeOps will be of utmost interest to the developers of shape analysis frameworks [12, 36, 60, 92, 103] and CUDA kernels for point cloud convolution [35, 102, 119]. Future developments may also be relevant to natural language processing: transformer architectures and attention layers fit a similar design pattern, albeit in higher-dimensional spaces [100, 106, 113].

Table 4: Accelerating geometric primitives: average time per shape on the ShapeNet dataset, with clouds of N = 2,048 points in dimension D = 3.

| Primitive / Neighborhood | PyTorch 40-NN | JAX 40-NN | JAX Set radius | Ours 40-NN | Ours Set radius |
|---|---|---|---|---|---|
| Local mean vectors | 686 μs | 1,052 μs | 469 μs | 121 μs | **12 μs** |
| Local covariance matrices | 700 μs | 1,093 μs | 1,259 μs | 138 μs | **23 μs** |
| MLP features (H = O = 8) | 737 μs | 1,180 μs | 4,089 μs | 192 μs | **75 μs** |
| MLP features (H = O = 16) | 775 μs | 1,253 μs | 7,043 μs | **240 μs** | 649 μs |
| Chamfer loss | 374 μs | 130 μs | | **21 μs** | |
| Energy distance | 486 μs | 378 μs | | **31 μs** | |

### 5.4 Easing the development of complex geometric programs

**Dimension reduction and geometric embeddings.** The uniform manifold approximation and projection (UMAP) algorithm [79] is a standard method for dimension reduction. It can be used with various input metrics and relies on the analysis of a KNN graph of the input data. To showcase the benefits of KeOps for data analysis, we benchmark three different implementations of this method: the reference CPU-only library [79]; CuML, a fast GPU implementation that relies on FAISS for nearest neighbor queries [65, 91]; and a custom CuML+KeOps implementation. As typical examples of UMAP for visualization, we embed several datasets in the Euclidean plane: the digits, SIFT and MNIST datasets (endowed with the Euclidean and Manhattan metrics); the Glove-25 dataset (cosine similarity); and the HyperE-10 and -50 embeddings (hyperbolic metric).

Since the initial construction of the KNN graph is the most compute-intensive part of the UMAP algorithm, results are similar to those of Section 5.2 (a full table of results is provided in the Supplementary Materials). KeOps provides a $\times 2-5$ speed-up for the visualization of low-dimensional datasets, but is outperformed by CuML+FAISS when the input dimension D exceeds 50. Crucially, we provide the only GPU method that can handle non-Euclidean metrics: for hyperbolic embeddings, our implementation is $\times 200$ faster than the baseline on the HyperE-10 and -50 datasets [72, 83].

**Optimal transport.** Optimal transport (OT) is a generalization of sorting to spaces of dimension $D > 1$ [81, 107]. It revolves around the resolution of a linear optimization problem whose value is usually known as the Wasserstein or Earth Mover's distance between two point clouds $(x_i)$ and $(y_j)$ [68, 93]. As detailed in [37, 89], modern OT solvers rely on iterated reductions performed on a cost matrix $C(x_i, y_j) = \|x_i - y_j\|$ or $\frac{1}{2}\|x_i - y_j\|^2$ and stand to benefit greatly from our library. To demonstrate this, we benchmark an exact linear solver [13, 42] against several variants and implementations of the Sinkhorn algorithm: a vanilla Sinkhorn loop [27, 29, 34, 41, 44, 94, 98, 112, 117]; an accelerated algorithm with simulated annealing [71]; and a fast multiscale scheme with kernel truncation [10, 96]. For the sake of numerical stability, we use symmetric updates [70] and perform all computations in the logarithmic domain [23]. To reflect the varied use cases of OT in the machine learning literature, we tackle two different problems: a high-precision matching between deformations of the Stanford dragon in dimension $D = 3$, and a low-precision matching between deformations of the Glove-25 dataset in dimension $D = 25$. These two regimes correspond to typical applications in computer vision [24, 38] and statistics [43, 46, 48], respectively.

A table of results is provided in the Supplementary Materials. In all settings, the switch from a PyTorch implementation to KeOps provides a $\times 5-20$ speed-up with no loss of precision. Most importantly, the low memory footprint of symbolic `LazyTensors` allows us to scale up to large problems ($N, M > 100k$) without memory overflows. Our support for block-wise sparsity masks also lets us provide the first implementation of a multiscale solver for discrete OT on the GPU. In practice, this means that large OT problems with $N = 100k$ or $1M$ samples can now be solved with high precision in fractions of a second instead of minutes or hours [37]. This implementation is straightforward to generalize to stochastic settings [2, 3, 47] and non-Euclidean cost functions [9, 28, 30]: we believe that it will open new ranges of applications to the OT community [37, 39].

## 6  Conclusion

KeOps combines a user-friendly interface with a high level of performance: we believe that it fills an important niche in the machine learning toolbox. We hope that our library will stimulate research in the field, with a simple but powerful structure that makes it a good tool for research off the beaten track. We look forward to feedback from users and keep the door open for contributors.

In months to come, our priority will be to improve performances on high-dimensional vectors with the newly released GPU Tensor cores and add support for quantization or approximation schemes such as the Nyström and FFM methods [5, 111]. We also work towards easing the deployment of pre-compiled binaries and target support of the ONNX standard [6]. These improvements should allow KeOps to become a standard toolbox in the field, both as an efficient backend for high-level software [12, 40, 45, 80] and as a versatile prototyping tool for theoretical research.

## Broader Impact

Our work targets a wide range of machine learning applications, from kernel methods to geometric deep learning. In these fields, our library lowers the barrier of entry to state-of-the-art performances: fast nearest neighbors queries or point cloud convolutions can now be implemented by researchers who have no background in parallel computing. We hope that this will empower small research teams and organizations who don't have access to dedicated teams of software engineers. More specifically, the flexibility of our library is ideally suited to the formulation of data-driven models for shape analysis and point cloud processing. Progress in these sectors can have a major impact in computer vision and medical imaging – topics that carry both risks and promises for society as a whole. We hope that our library will promote the growth of a diverse ecosystem of academic and industrial actors, and look forward to seeing applications of our work to e.g. computational anatomy.

## Acknowledgments and Disclosure of Funding

The three first authors are the project leaders: they contributed equally to the library and its documentation. Michael Bronstein is supported by the ERC Consolidator grant No. 724228 (LEMAN). KeOps was first motivated by applications to computational anatomy, in collaboration with Alain Trouvé. We also thank Ghislain Durif, who develops the R bindings of KeOps, as well as François-David Collin, who helps us to maintain our hardware and testing infrastructure.

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
