[Supplementary Material]

# Extended benchmarks and implementations

**Content of the supplementary materials.** Our library is freely available on the PyPi (`pip install pykeops`) and CRAN (`install.packages("rkeops")`) repositories. Its user interface and inner workings are fully documented on our website (`www.kernel-operations.io`), with source code available under the permissive MIT license (`www.github.com/getkeops/keops`).

The full codes for our benchmarks have been integrated to our documentation as tutorials. In the pages below, we present supporting material for the discussion of Section 5: we include all the relevant equations, code samples and tables of results. The hardware configuration and datasets are described at the start of Section 5.

## A Kernel methods

**Kernels, Gaussian process regression.** As discussed in Section 5.1, KeOps is ideally suited to the implementation of kernel methods: LazyTensors can be used to represent arbitrary kernel matrices with a low memory footprint and high performance. As an example, we show how to interface our library with the standard solvers of the `scipy.sparse.linalg` package [66] – a reference toolbox for e.g. the computation of Laplacian eigenvectors on graphs and meshes.

**SciPy interface.** If $x = (x_i) \in \mathbb{R}^{N \times D}$ is a cloud of N points in $\mathbb{R}^D$ and if $b = (b_i) \in \mathbb{R}^{N \times E}$ is a signal of dimension E supported by the $x_i$'s, the code below implements a fast conjugate gradient solver for the resolution of a linear system with respect to $a = (a_i) \in \mathbb{R}^{N \times E}$:

$$b = (\alpha \operatorname{Id} + K_{x,x}) a \qquad \text{i.e.} \qquad a = (\alpha \operatorname{Id} + K_{x,x})^{-1} b, \qquad (3)$$

where $K_{x,x} = (K_{x_i,x_j}) = (\exp(-\|x_i - x_j\|/\sigma)$ is a $(N, N)$ symmetric positive definite matrix associated to an exponential kernel of radius $\sigma > 0$ and where $\alpha \geqslant 0$ is the strength of a $L^2$-Tikhonov regularization. This operation is a the heart of Gaussian process regression [57]: it is usually known as Kriging in geostatistics, kernel regression in data sciences or spline regression in imaging. We illustrate some typical use cases in Figure 3.

```python
import numpy as np  # NumPy arrays on the CPU
from scipy.sparse import diags  # Sparse diagonal matrices
from scipy.sparse.linalg import aslinearoperator, cg  # Conjugate gradient
from pykeops.numpy import LazyTensor  # Symbolic wrapper for NumPy arrays

# Toy problem in dimension D = 50:
N, D, E = 10 ** 6, 50, 1  # samples, features, signals
x = np.random.randn(N, D).astype('float32')  # float16, float32 and float64
b = np.random.randn(N, E).astype('float32')  # are all supported by KeOps.

sigma = .2  # radius of the exponential kernel
alpha = .5  # ridge/Tikhonov regularization

# Build the symbolic (N, N) kernel matrix:
x_i = LazyTensor(x.reshape(N, 1, D))       # (N, 1, D) data samples
x_j = LazyTensor(x.reshape(1, N, D))       # (1, N, D) data samples

D_ij = ((x_i - x_j) ** 2).sum(2).sqrt()  # (N, N) distances
K_ij = (- D_ij / sigma).exp()            # (N, N) exponential kernel matrix

# Turn the LazyTensor into a SciPy object, add Tikhonov regularization:
K = aslinearoperator(K_ij)  # Transparent duck typing from KeOps to SciPy
Ka = K + aslinearoperator(diags(alpha * np.ones(N)))  # Standard SciPy syntax
Ka.dtype = np.dtype('float32')  # Use the correct precision

# Interface KeOps with all the standard solvers of scipy.sparse.linalg:
a = cg(Ka, b)  # Conjugate gradient: eigenproblem solvers, etc. are also supported.
```

<div style="text-align:center">(a) Kernel regression in 1D.        (b) Kriging in 2D.</div>

Figure 3: **Kriging**, also known as **kernel**, **spline** or **Gaussian process regression** is a fundamental tool in data sciences that relies on the resolution of large kernel linear systems (3). Out-of-the-box, our symbolic tensors can be interfaced with standard libraries for linear algebra such as SciPy [66]. This lets users scale up standard iterative solvers to datasets of N = 10k to 10M samples in seconds or minutes. (a) As a first example, we work with a Gaussian kernel $k(x, y) = \exp(-|x - y|^2/2\sigma^2)$ of deviation $\sigma = 0.1$ on the real line. We use N =10k samples in dimension D = 1, with a scalar-valued signal (E = 1): we represent the data with blue points $(x_i, b_i)$ in the graph. The red curve corresponds to the kernel regression $x \mapsto \sum_{i=1}^{N} k(x, x_i) a_i$, a smooth curve that does not overfit to noise thanks to the Tikhonov regularization. (b) The second example is representative of applications to geostatistics: we work with N =10k samples in dimension D = 2. Thanks to an exponential kernel of deviation $\sigma = 0.1$ and a small amount of Tikhonov regularization, we retrieve a continuous interpolation of a noisy scalar signal (E = 1) on the whole domain: a plausible terrain model, displayed as an image in the background while every point corresponds to a sample $x_i$ with color $b_i$. The flexible structure of our library empowers researchers, who can use LazyTensors to perform fast kernel regressions on arbitrary domains, such as the sphere or the Poincaré plane.

**Performance.**    Providing rigorous and precise benchmarks for iterative linear solvers is an arduous task: a wide range of methods have been proposed to accelerate the resolution of systems that involve e.g. smooth kernel functions. Depending on their specific needs, users often have to pick a method and parameter values that reach a satisfying trade-off between speed and accuracy.

Nevertheless, according to our experiments with default precision settings, N = 1k to 10M points in dimension D = 1 to 100 and varied kernel functions (Gaussian, exponential, Cauchy, etc.), we observe that SciPy+KeOps implementations are consistently ×10-50 times faster than their standard PyTorch counterparts (`torch.solve(...)`) and ×1,000-5,000 times faster than a vanilla resolution with SciPy on the CPU. These speed-ups come from our efficient use of CUDA registers and could be applied to accelerate most large-scale solvers in the field [57]: we believe that our library will be of interest to many researchers who work with Gaussian processes or kernel matrices.

# B    Clustering

## B.1    K-Means: Lloyd's algorithm

**Fast clustering with K-means.**    We now discuss the applications of symbolic tensors to clustering. We first consider the problem of partitioning a dataset $(x_i) \in \mathbb{R}^{N \times D}$ of N points in $\mathbb{R}^D$ in K distinct clusters. The K-means method or (discrete) "Lloyd's algorithm" is probably the most common approach to the question: we work with a collection $(c_k) \in \mathbb{R}^{K \times D}$ of K cluster "centroids" in $\mathbb{R}^D$, class labels $(l_i) \in [\![1, K]\!]^N$ for every point $x_i$ and update both parameters alternatively to minimize

the within-cluster sum of squared distances:

$$\text{SSD}(c_k, l_i) \;=\; \sum_{k=1}^{K} \sum_{l_i=k} \| x_i - c_k \|^2. \tag{4}$$

At every iteration of the K-means loop, we first assign each point $x_i$ to the closest centroid $c_k$ (i.e. minimize SSD with respect to the $l_i$'s) before updating $c_k$ as the mean of all points $x_i$ such that $l_i = k$. Using KeOps for the assignment step, we can write a fast and simple implementation of this algorithm as follows:

```python
def KMeans(x, K, niter, verbose=True):
    """
    points  ->  labels, centroids
    (N, D)  ->   (N,),   (K, D)
    """
    N, D = x.shape # Number of samples, dimension of the ambient space

    c = x[:K, :].clone() # Simple initialization for the centroids
    # Encoding as symbolic tensors:
    x_i = LazyTensor(x.view(N, 1, D)) # (N, 1, D) symbolic tensor
    c_j = LazyTensor(c.view(1, K, D)) # (1, K, D) symbolic tensor

    for _ in range(niter):   # K-means loop
        # Assignment step:
        D_ij = ((x_i - c_j) ** 2).sum(-1) # (N, K) squared distances
        l = D_ij.argmin(dim=1).long().view(-1) # Points -> Nearest cluster

        # Compute the cluster mean values:
        weights = torch.bincount(l).type_as(x)
        for d in range(D):   # In-place update of the centroids:
            c[:, d] = torch.bincount(l, weights=x[:, d]) / weights

    return l, c  # Labels, centroids
```

Note that we use a weighted `torch.bincount` method for the update step, which avoids looping over the class index in $[\![1, K]\!]$. In practice, this second step relies on scattered memory accesses and is the bottleneck of the K-means loop for small-scale problems. On our system, with N = 1M, K = 1k and D = 100, this implementation performs 10 iterations in less than a second (0.81s on average).

**Manhattan distance.** Our library is versatile, and lets users prototype arbitrary generalizations of standard algorithms. For instance, we can easily implement an $L^1$-Manhattan variant of Lloyd's algorithm to minimize the robust cost function:

$$\text{SD}_{L^1}(c_k, l_i) \;=\; \sum_{k=1}^{K} \sum_{l_i=k} \| x_i - c_k \|_1 \;=\; \sum_{k=1}^{K} \sum_{l_i=k} \sum_{d=1}^{D} | \, x_i[d] - c_k[d] \, |. \tag{5}$$

In the assignment step, we replace the squared Euclidean norms by Manhattan distances, prior to the nearest neighbour search:

```python
        D_ij = ((x_i - c_j).abs()).sum(-1) # (N, K) Manhattan distances
```

As for the update step, we replace means by medians to compute the new centroids $c_k$:

```python
        # Update cluster centroids:
        for k in range(K):
            c[k,:] = torch.median(x[l==k,:], dim=0)[0]
```

Note that in this case, there is no simple way of avoiding a loop over K for the update step with PyTorch. As a consequence, the performance of this implementation drops to an average of 1.85s for the same test dimensions – N = 1M, K = 1k, D = 100 and 10 iterations.

(a) Lloyd's algorithm.

(b) Lloyd's algorithm, $L^1$ variant.

(c) GMM with diagonal covariances.

(d) GMM with full covariances.

Figure 4: Clustering of a synthetic 2D dataset (N = 10k, D = 2), into K = 5 classes with four different methods. (a) The standard Lloyd's algorithm for the $L^2$-Euclidean metric. (b) A variant of Lloyd's algorithm for the $L^1$-Manhattan metric. (c) The EM algorithm on a Gaussian mixture model with diagonal covariances. (d) The EM algorithm on a Gaussian mixture model with full covariances. We display the points $x_i$ in the unit square, colored according to the class labels $l_i$. For the Gaussian mixture models (c-d), we also display the model likelihood (6) in the background, with colors that reflect the dominant cluster at any given location. All the experiments were performed using KeOps, following the implementations of Section B.

## B.2 Gaussian mixture models: the EM algorithm

**Notations.** We now detail the content of Table 1. We consider a dataset $(x_i) \in \mathbb{R}^{N \times D}$ of N points in $\mathbb{R}^D$ and fit a Gaussian mixture model $\mathrm{GMM}(w_j, \mu_j, \Sigma_j; j \in [\![1, K]\!])$ that is parameterized by a collection of K:

1. **weights** $w_j \geqslant 0$ that sum up to 1,
2. **mean values** $\mu_j \in \mathbb{R}^D$,
3. **covariance matrices** $\Sigma_j \in \mathbb{R}^{D \times D}$.

The likelihood of the model at any point $x \in \mathbb{R}^D$ is given by:

$$\mathrm{likelihood}(x) = \sum_{j=1}^{K} \frac{w_j}{(2\pi)^{D/2}\sqrt{\det(\Sigma_j)}} \exp\left(-\tfrac{1}{2}(x - \mu_j)^\top \Sigma_j^{-1}(x - \mu_j)\right). \tag{6}$$

**EM iterations.**  Starting from a random initialization, we fit the model to the data using the standard Expectation-Maximization algorithm. Its iterations read as follows:

1. **E-step:** compute membership probabilities. For every point $x_i$ and component $(w_j, \mu_j, \Sigma_j)$, we compute the likelihood ratio:

$$\pi_{i,j} \;=\; \frac{\text{likelihood}_{j\text{-th component}}(x_i)}{\text{likelihood}_{\text{full}}(x_i)} \tag{7}$$

$$= \frac{w_j \exp\left(-\frac{1}{2}(x_i - \mu_j)^\top \Sigma_j^{-1}(x_i - \mu_j)\right)/\sqrt{\det(\Sigma_j)}}{\sum_{k=1}^{K} w_k \exp\left(-\frac{1}{2}(x_i - \mu_k)^\top \Sigma_k^{-1}(x_i - \mu_k)\right)/\sqrt{\det(\Sigma_k)}} \tag{8}$$

$(\pi_{i,j}) \in \mathbb{R}_{\geqslant 0}^{N \times K}$ is encoded as a $(N, K)$ array whose lines sum up to 1.

2. **M-step:** update the model parameters. We execute sequentially the following equations:

$$P_j \;\leftarrow\; \sum_{i=1}^{N} \pi_{i,j}\,, \qquad\qquad w_j \;\leftarrow\; P_j/N\,, \tag{9}$$

$$\mu_j \;\leftarrow\; \frac{1}{P_j} \sum_{i=1}^{N} \pi_{i,j} x_i\,, \qquad \Sigma_j \;\leftarrow\; \frac{1}{P_j} \sum_{i=1}^{N} \pi_{i,j}(x_i - \mu_j)(x_i - \mu_j)^\top\,. \tag{10}$$

For the sake of numerical stability, we add a small value $\varepsilon = 10^{-7}$ to the class scores $P_j$ when they are used as denominators. Alternatively, we could work with their logarithms and stabilized log-sum-exp reductions: these are fully supported by our library.

Inverting the K covariance matrices $\Sigma_j$ to compute the precisions $\Sigma_j^{-1} \in \mathbb{R}^{D \times D}$ for every E-step can be costly. In Table 1, we also benchmark an alternative version of the algorithm where the covariances are assumed to be diagonal matrices and encoded as positive vectors $\sigma_j \in \mathbb{R}^D$.

```python
# Input:  points is (N, D)
# Params: weights is (K,), means is (K, D), covariances is (K, D, D)

for _ in range(niter):
    # Expectation step: compute membership probabilities ------------------
    # Compute mixture weights:
    precisions = covariances.inverse()   # (K, D, D)
    w = weights * torch.sqrt(precisions.det())   # (K,)

    # Encoding as symbolic tensors:
    x_i = LazyTensor(points.view(N, 1, D))    # (N, 1, D)
    m_j = LazyTensor( means.view(1, K, D))    # (1, K, D)
    w_j = LazyTensor(w.view(1, K, 1))         # (1, K, 1)

    # Gaussian likelihoods:
    P_j = LazyTensor(precisions.reshape(1, K, D * D))   # (1, K, D*D)
    D_ij = ((x_i - m_j) * P_j.matvecmult(x_i - m_j)).sum(dim=2)   # (N, K)
    K_ij = (- D_ij / 2).exp() * w_j   # (N, K)

    # Bayes normalization constant:
    BN = K_ij.sum(dim=1)   # (N,)
    BN_i = LazyTensor(BN.view(N, 1, 1) + eps)   # (N, 1)

    # Compute the membership probabilities:
    P_ij = K_ij / BN_i    # (N, K)

    # Maximization step: update the mixture parameters -------------------
    P = P_ij.sum(dim=0)         # (K, 1)
    weights = P.view(-1) / N    # (K,)
    means = (P_ij * x_i).sum(dim=0) / (P + eps)   # (K, D)

    # New means to compute the adjusted covariances:
    m_j = LazyTensor(means.view(1, K, D))   # (1, K, D)

    # Covariance matrices
    covariances = (P_ij * (x_i-m_j).tensorprod(x_i-m_j)).sum(0).view(K,D,D)
    covariances = covariances / (P.view(K, 1, 1) + eps)   # (K, D, D)
```

## C   Dimensionality reduction

As discussed in Section 5.3, we benchmark: the original UMAP implementation on the CPU; the CuML+FAISS implementation on the GPU; a CUML+KeOps pipeline which relies on symbolic tensors to build the KNN graph of a dataset before relying on CuML to construct the low-dimensional embedding. This pipeline allows us to compute UMAP embeddings with arbitrary metrics on the input datasets: it is indicative of the versatility of KeOps, which can be interfaced with a wide range of standard libraries. Results are presented in Table 5, with examples of embeddings shown in Figure 5.

**Note on the hyperbolic metric.**   The HyperE-10 and -50 datasets provide reference embeddings of real world data into hyperbolic spaces of dimensions 10 and 50. In practice, the datasets both rely on the Poincaré ball model and provide a scaling factor that should be used to recover the hyperbolic distance between any two vectors. If $x_i$ and $x_j$ are two samples in the dataset, encoded as vectors of norms $\|x_i\|, \|x_j\| < 1$ in $\mathbb{R}^D$, the hyperbolic distance between them is given by:

$$ \mathrm{d}(x_i, x_j) \; = \; \mathrm{arccosh}\left(1 + 2\,\frac{\|x_i - x_j\|^2}{(1 - \|x_i\|^2)(1 - \|x_j\|^2)}\right) \; / \; \text{ScalingFactor} \,, \tag{11} $$

where $\|\cdot\|$ denotes the standard Euclidean norm in $\mathbb{R}^D$. In order to perform a KNN search efficiently, we remark that $x \mapsto \mathrm{arccosh}(1 + 2x)$ is an increasing mapping. Since the values of $(1 - \|x_i\|^2)$ can be computed ahead of the KNN reduction, we can build our KNN graph with:

```
1   # x is a (N, D) array with double precision. We first compute the scaling factors:
2   u = 1. / (1. - (x ** 2).sum(dim=1))   # With double = float64 precision.
3   x, u = x.float(), u.float()   # We can use float32 precision after this step.
4
5   # And encode our variables as symbolic tensors:
6   x_i = LazyTensor(x.view(N, 1, D))
7   x_j = LazyTensor(x.view(1, N, D))
8   u_i = LazyTensor(u.view(N, 1, 1))
9   u_j = LazyTensor(u.view(1, N, 1))
10
11  # We can then perform the KNN search efficiently:
12  D_ij = ((x_i - x_j) ** 2).sum(dim=2) * u_i * u_j
13  distances, indices = D_ij.Kmin_argKmin(K, dim = 1)
14
15  # And compute the genuine hyperbolic distances to the K-nearest neighbors:
16  acosh = lambda x : torch.log( x + (x ** 2 - 1.) ** 0.5)
17  distances = arccosh(1. + 2 * distances) / scaling_factor
```

Table 5: Dimension reduction using the UMAP algorithm. We record the time to embed datasets in the Euclidean plane. When the input metric is Euclidean, the dataset is first pre-processed with a PCA as advised by the UMAP documentation: we keep $95\%$ of the total variance.

| Dataset | Metric | N | D | PCA preprocessing | Umap | CuML | CuML+**Ours** |
|---------|--------|-----|-----|-------------------|------|------|---------------|
| Digits | $L^2$ | 1.8k | 64 | 4 ms $\to D' = 28$ | 5.8 s | 170 ms | **32 ms** |
| MNIST | $L^2$ | 60k | 784 | 68 ms $\to D' = 153$ | 38 s | **450 ms** | 670 ms |
| MNIST | $L^1$ | 60k | 784 | —- | 43 s | —- | **2.3 s** |
| SIFT | $L^2$ | 1M | 128 | 64 ms $\to D' = 71$ | 1,380 s | **28 s** | 53 s |
| GloVe-25 | $\langle,\rangle$ | 1.2M | 25 | —- | 1,660 s | 31 s | **29 s** |
| HyperE-10 | $\mathbb{H}^D$ | 105k | 10 | —- | 150 s | —- | **560 ms** |
| HyperE-50 | $\mathbb{H}^D$ | 105k | 50 | —- | 200 s | —- | **900 ms** |

Figure 5: UMAP embeddings into the Euclidean plane. Top: MNIST dataset with a Manhattan input metric, colored by label. Bottom: HyperE-50 (WordNet) dataset with a hyperbolic input metric.

# D Geometric primitives

**Weighted average on a neighborhood.** We now detail the results of Table 4. As described in Section 5.3, we consider batches of B point clouds $(x_i) \in \mathbb{R}^{N \times D}$, with N = 2,048 and D = 3. Depending on the memory footprint of the computations, B is equal to 1, 10 or 100 and ensures that GPU cores are used efficiently: batches of point clouds are encoded as large (B, N, D) arrays. Following standard procedure for point cloud processing, we compute local features as:

$$a_i \leftarrow \frac{\sum_{j=1}^{M} w(x_i, x_j) F(x_i, x_j)}{\sum_{j=1}^{M} w(x_i, x_j)}, \quad \forall i \in [\![1, N]\!] \tag{12}$$

where $w(x_i, x_j) \geqslant 0$ is a weight on the interaction $(x_i, x_j)$ and $F$ is a vector-valued function.

In our benchmarks, we consider two types of weight functions $w$:

1. A KNN window $w(x_i, x_j)$ which is equal to 1 if $x_j$ is one of the K = 40 nearest neighbors of $x_i$ and 0 otherwise. It is implemented using a batched KNN search in dimension D = 3 and advanced indexing operators. Just as in Table 3, we use KeOps to accelerate the construction of the KNN graph and otherwise rely on standard PyTorch syntax to build up local neighborhoods as (B, N, K, D) arrays.

2. A Gaussian window of radius $\sigma > 0$:

$$w(x_i, x_j) = \exp\left(-\|x_i - x_j\|^2 / 2\sigma^2\right). \tag{13}$$

It is implemented using symbolic operations, as detailed below.

**Local mean.** In our first example, we compute the local average:

$$\mu_i \leftarrow \frac{\sum_{j=1}^{M} w(x_i, x_j) x_j}{\sum_{j=1}^{M} w(x_i, x_j)} \in \mathbb{R}^D, \quad \forall i \in [\![1, N]\!]. \tag{14}$$

In the code below, we compute both the numerator and denominator in one pass through the data. The trick is to append a "1" to the feature vectors $x_j$ in order to retrieve both:

$$w_i \leftarrow \sum_{j=1}^{M} w(x_i, x_j) \cdot 1 \qquad \text{and} \qquad m_i \leftarrow \sum_{j=1}^{M} w(x_i, x_j) x_j \tag{15}$$

with a single reduction call.

```
1   def local_mean(points, radius):
2       """
3        points,  radius ->   means
4       (B, N, D),    1   -> (B, N, D)
5       """
6       B, N, D = point.shape  # Batch-size, number of points, features
7       points = points / radius  # Normalize the window size to 1
8
9       # Add a "1" at the start of every vector, retrieve a (B, N, D+1) array:
10      x = torch.cat((torch.ones_like(points[:,:,:1]), points), dim = -1)
11
12      # Encode as symbolic tensors:
13      x_i = LazyTensor(x.view(B, N, 1, D+1))  # (B, N, 1, D+1)
14      x_j = LazyTensor(x.view(B, 1, N, D+1))  # (B, 1, N, D+1)
15
16      # Neighborhood window - a Gaussian function:
17      D_ij = ((x_i - x_j) ** 2).sum(-1)  # (B, N, N), squared distances
18      K_ij = (- D_ij / 2).exp()          # (B, N, N), Gaussian kernel
19
20      # Local sum:
21      M_ij = K_ij * x_j  # (B, N, N, D+1)
22      M_i  = M_ij.sum(dim = 2)  # (B, N, D+1) : weights and sums
23
24      # Normalize by the sum of the weights:
25      w_i = M_i[:,:,:1]  # (B, N, 1)
26      m_i = M_i[:,:,1:]  # (B, N, D)
27      return radius * m_i / w_i  # (B, N, D)
```

**Local covariance.** In our second example, we compute the local covariance matrices:

$$\Sigma_i \leftarrow \frac{\sum_{j=1}^{M} w(x_i, x_j)\,(x_j - \mu_i)(x_j - \mu_i)^\top}{\sum_{j=1}^{M} w(x_i, x_j)} \ \in \mathbb{R}^{D \times D}, \ \ \forall i \in [\![1, N]\!]\,, \qquad (16)$$

where $\mu_i$ is defined as in (14). Using standard identities, we can rewrite this local descriptor as:

$$\Sigma_i \leftarrow \frac{1}{w_i}\Big(c_i - \frac{1}{w_i} m_i m_i^\top\Big)\,, \qquad (17)$$

where:

$$w_i \leftarrow \sum_{j=1}^{M} w(x_i, x_j)\,, \qquad (18)$$

$$m_i \leftarrow \sum_{j=1}^{M} w(x_i, x_j)\, x_j\,, \qquad (19)$$

$$c_i \leftarrow \sum_{j=1}^{M} w(x_i, x_j)\, x_j x_j^\top\,. \qquad (20)$$

For optimal performances, we rely on the same trick as in (15) to compute all these quantities in one pass through the data. We append a "1" at the start of every vector $x_j$ and compute:

$$\left[\begin{array}{c|c} w_i & m_i \\ \hline m_i^\top & c_i \end{array}\right] \ = \ C_i \leftarrow \sum_{j=1}^{M} w(x_i, x_j)\,[1, x_j][1, x_j]^\top \qquad (21)$$

```python
def local_covariance(points, radius):
    """

    points,  radius ->  covariances
    (B, N, D),    1    ->  (B, N, D, D)
    """
    B, N, D = point.shape  # Batch-size, number of points, features
    points = points / radius  # Normalize the window size to 1

    # Add a "1" at the start of every vector, retrieve a (B, N, D+1) array:
    x = torch.cat((torch.ones_like(points[:,:,:1]), points), dim=-1)   # (B, N, D+1)

    # Encode as symbolic tensors:
    x_i = LazyTensor(x[:,:,None,:])   # (B, N, 1, D+1)
    x_j = LazyTensor(x[:,None,:,:])   # (B, 1, N, D+1)

    # Neighborhood window - a Gaussian function:
    D_ij = ((x_i - x_j) ** 2).sum(-1)   # (B, N, N), squared distances
    K_ij = (- D_ij / 2).exp()           # (B, N, N), Gaussian kernel

    # Local sum - compute descriptors of order 0, 1 and 2:
    C_ij = (K_ij * x_j).tensorprod(x_j)   # (B, N, N, (D+1)*(D+1))
    C_i  = C_ij.sum(dim = 2).view(B, N, D+1, D+1)   # (B, N, D+1, D+1)

    # Extract local descriptors of order 0, 1 and 2:
    w_i = C_i[:,:,:1,:1]                  # (B, N, 1, 1), weights
    m_i = C_i[:,:,:1,1:] *  radius        # (B, N, 1, D), sum
    c_i = C_i[:,:,1:,1:] * (radius**2)    # (B, N, D, D), outer products

    # Compute the covariance matrix:
    cov_i = (c_i - (m_i.transpose(3, 2) * m_i) / w_i) / w_i   # (B, N, D, D)
    return cov_i
```

**MLP features.** Going further, we show how to use our library to compute neural features. Following standard practice in geometric deep learning, we rely on a multi-layer perceptron:

$$F : x \in \mathbb{R}^D \mapsto A_2 \operatorname{ReLU}(A_1 x + b_1) + b_2 \in \mathbb{R}^O \tag{22}$$

parameterized by weight matrices $A_1 \in \mathbb{R}^{H \times D}$, $A_2 \in \mathbb{R}^{O \times H}$ and bias vectors $b_1 \in \mathbb{R}^H$, $b_2 \in \mathbb{R}^O$. "ReLU" denotes the rectified linear unit, or positive part, applied coordinate-wise on vectors of $\mathbb{R}^H$. For the sake of simplicity, we compute the MLP correlations:

$$a_i \leftarrow \sum_{j=1}^{M} w(x_i, x_j) F(x_j - x_i) \tag{23}$$

$$= \sum_{j=1}^{M} w(x_i, x_j) (A_2 \operatorname{ReLU}(f_j - f_i + b_1) + b_2), \tag{24}$$

where the hidden features $f_i = A_1 x_i$ are computed ahead of the sum reduction. We stress that the code below is fully differentiable: gradients can be computed with respect to all parameters.

We note that the matrix-vector product with $A_2$ is a $\mathcal{O}(OH)$ operation. In practice, our bruteforce CUDA engine is most efficient if the product $O \cdot H$ is smaller than 100: beyond this threshold, performance decrease sharply as in e.g. the fourth line of Table 4. KNN implementations are ideally suited to the computation of localized but complex features, whereas symbolic matrices let us compute efficiently simple descriptors at all scales.

```
1  def MLP_features(points, A_1, B_1, A_2, B_2, radius):
2      """
3       points,  weights_1, bias_1, weights_2, bias_2, radius  ->  features
4       (B, N, D), (H, D),   (H,),    (O, H),    (O,),    1     ->  (B, N, O)
5      """
6      B, N, D = points.shape
7      x = points / radius  # Normalize the window size to 1
8
9      # Apply the first linear operator on the features:
10     f = points @ A_1.t()  # (B, N, H)
11
12     # Encode the variables as symbolic tensors:
13     # Positions:
14     x_i = LazyTensor(x.view(B, N, 1, D))   # (B, N, 1, D)
15     x_j = LazyTensor(x.view(B, 1, N, D))   # (B, 1, N, D)
16     # Features:
17     f_i = LazyTensor(f.view(B, N, 1, -1))  # (B, N, 1, H)
18     f_j = LazyTensor(f.view(B, 1, N, -1))  # (B, 1, N, H)
19     # MLP parameters:
20     b_1 = LazyTensor(B_1.view(1, 1, 1, -1))  # (1, 1, 1, H)
21     a_2 = LazyTensor(A_2.view(1, 1, 1, -1))  # (1, 1, 1, O * H)
22     b_2 = LazyTensor(B_2.view(1, 1, 1, -1))  # (1, 1, 1, O)
23
24     # Compute the MLP values:
25     M_ij = (f_j - f_i + b_1).relu()     # (B, N, N, H)
26     M_ij = a_2.matvecmult(M_ij) + b_2   # (B, N, N, O)
27
28     # Neighborhood window - a Gaussian function:
29     D_ij = ((x_i - x_j) ** 2).sum(-1)   # (B, N, N), squared distances
30     K_ij = (- D_ij / 2).exp()   # (B, N, N), Gaussian kernel
31
32     # Sum on the neighborhood:
33     C_ij = K_ij * M_ij   # (B, N, N, O)
34     features  = C_ij.sum(dim = 2)   # (B, N, O)
35
36     return features
```

**Chamfer loss.** Beyond geometric descriptors, symbolic tensors let us work efficiently with global, geometric loss functions. If $(x_i) \in \mathbb{R}^{N \times D}$ and $(y_j) \in \mathbb{R}^{M \times D}$ are two clouds of N and M points in $\mathbb{R}^D$, the "chamfer" or "soft-Hausdorff" loss between them reads:

$$\text{Chamfer}(x_i, y_j) \;=\; \frac{1}{2N} \sum_{i=1}^{N} \min_{j=1}^{M} \|x_i - y_j\| \;+\; \frac{1}{2M} \sum_{j=1}^{M} \min_{i=1}^{N} \|x_i - y_j\|. \tag{25}$$

Variants of this formula are used, for instance, in the Iterative Closest Point (ICP) algorithm. We leverage our fast nearest neighbor finder to implement it as follows:

```
def squared_distances(x, y):
    """
     source,   target   -> squared distances
    (B, N, D), (B, M, D) ->    (B, N, M)
    """
    B, N, D = x.shape  # Batch size, number of source points, features
    _, M, _ = y.shape  # Batch size, number of target points, features

    # Encode as symbolic tensors:
    x_i = LazyTensor(x.view(B, N, 1, D))  # (B, N, 1, D)
    y_j = LazyTensor(y.view(B, 1, M, D))  # (B, 1, M, D)

    # Symbolic matrix of squared distances:
    D_ij = ((x_i - y_j)**2).sum(-1)     # (B, N, M), squared distances
    return D_ij

def chamfer_loss(x, y):
    """
     source,   target   -> loss values
    (B, N, D), (B, M, D) ->    (B,)
    """
    D_ij = squared_distances(x, y)   # (B, N, M) symbolic matrix
    D_xy = D_ij.min(dim=2).sqrt()    # (B, N), distances from x to y
    D_yx = D_ij.min(dim=1).sqrt()    # (B, M), distances from y to x
    return (D_xy.mean(dim=1) + D_yx.mean(dim=1)).view(-1) / 2   # (B,)
```

**Energy distance.** Going further, we can combine symbolic tensors and sum reductions to compute generic kernel norms, which produce smoother gradients for e.g. shape registration. These quantities are also known as Maximum Mean Discrepancies (MMDs) in statistics or generalized electrostatic energies in physics. As an example, the code below implements the Energy Distance [99]:

$$\begin{aligned}\text{ED}(x_i, y_j) \;=\;& \frac{1}{NM} \sum_{i=1}^{N} \sum_{j=1}^{M} \|x_i - y_j\| \\ &- \frac{1}{2N^2} \sum_{i=1}^{N} \sum_{j=1}^{N} \|x_i - x_j\| \;-\; \frac{1}{2M^2} \sum_{i=1}^{M} \sum_{j=1}^{M} \|y_i - y_j\|.\end{aligned} \tag{26}$$

```
def energy_distance(x, y):
    """
     source,   target   -> loss values
    (B, N, D), (B, M, D) ->    (B,)
    """
    N, M = x.shape[1], y.shape[1]   # Numbers of source and target points

    D_xy = squared_distances(x, y).sqrt().sum(dim=2)   # (B, N), distances x<->y
    D_xx = squared_distances(x, x).sqrt().sum(dim=2)   # (B, N), distances x<->x
    D_yy = squared_distances(y, y).sqrt().sum(dim=2)   # (B, M), distances y<->y

    return (D_xy.sum(dim=1) /   (N*M)
            - D_xx.sum(dim=1) / (2*N*N)
            - D_yy.sum(dim=1) / (2*M*M)).view(-1)   # (B,)
```

# E  Optimal transport

**The optimal transport problem.**    As discussed in Section 5.3, optimal transport generalizes sorting to spaces of dimension $D > 1$. We now consider two point clouds $(x_i) \in \mathbb{R}^{N \times D}$, $(y_j) \in \mathbb{R}^{M \times D}$ with non-negative weights $(\alpha_i) \in \mathbb{R}_{\geqslant 0}^{N}$ and $(\beta_j) \in \mathbb{R}_{\geqslant 0}^{M}$ that sum up to 1. These arrays encode two discrete probability measures $\alpha$ and $\beta$ on $\mathbb{R}^D$, understood as weighted sums of Dirac masses $\delta_x$:

$$\alpha = \sum_{i=1}^{N} \alpha_i \delta_{x_i} \qquad \text{and} \qquad \beta = \sum_{j=1}^{M} \beta_j \delta_{y_j} . \tag{27}$$

If $\mathbf{C}(x_i, y_j)$ denotes an arbitrary cost function on $\mathbb{R}^D \times \mathbb{R}^D$, the optimal transport cost between the two discrete measures $\alpha$ and $\beta$ reads:

$$\mathrm{OT}(\alpha_i, x_i, \beta_j, y_j) = \min_{(\pi_{i,j}) \in \mathbb{R}_{\geqslant 0}^{N \times M}} \sum_{i=1}^{N} \sum_{j=1}^{M} \pi_{i,j} \, \mathbf{C}(x_i, y_j) \tag{28}$$

$$\text{subject to} \quad \forall i, j, \ \pi_{i,j} \geqslant 0, \quad (\pi \mathbf{1})_i = \sum_{j=1}^{M} \pi_{i,j} = \alpha_i, \quad (\pi^\top \mathbf{1})_j = \sum_{i=1}^{N} \pi_{i,j} = \beta_j .$$

The optimal transport plan $(\pi_{i,j})$ is a non-negative $(N, M)$ array whose lines sum up to $(\alpha_i)$ and whose columns sum up to $(\beta_j)$. In the remainder of this section, we use the quadratic cost $\mathbf{C}(x_i, y_j) = \frac{1}{2}\|x_i - y_j\|^2$: up to a factor $1/2$, the cost value $\mathrm{OT}(\alpha_i, x_i, \beta_j, y_j)$ is the squared Wasserstein-2 distance between $\alpha$ and $\beta$.

**Dual problem.**    A fundamental remark was made by Kantorovitch in [68]: the linear optimization problem (28) is equivalent to a simpler dual problem on variables of size $N$ and $M$:

$$\mathrm{OT}(\alpha_i, x_i, \beta_j, y_j) = \max_{\substack{(f_i) \in \mathbb{R}^N \\ (g_j) \in \mathbb{R}^M}} \sum_{i=1}^{N} \alpha_i f_i + \sum_{j=1}^{M} \beta_j g_j \ \text{ s.t. } \forall i, j, \ f_i + g_j \leqslant \mathbf{C}(x_i, y_j) . \tag{29}$$

The dual vectors $(f_i)$ and $(g_j)$ are unique up to an additional constant. They are often understood as the sampled values $f_i = f(x_i)$, $g_j = g(y_j)$ of continuous dual potentials on the input point clouds.

**Entropic regularization.**    Optimal transport solvers compute the optimal dual vectors $(f_i)$ and $(g_j)$ associated to any discrete input configuration $(\alpha_i, x_i, \beta_j, y_j)$. To this end, a common strategy is to add a small entropic barrier to the primal problem (28). If $\varepsilon > 0$ is a positive temperature, we can apply the Fenchel-Rockafellar theorem and write the regularized primal and dual problems as:

$$\mathrm{OT}_\varepsilon(\alpha_i, x_i, \beta_j, y_j) = \min_{(\pi_{i,j}) \in \mathbb{R}_{\geqslant 0}^{N \times M}} \sum_{i=1}^{N} \sum_{j=1}^{M} \pi_{i,j} \, \mathbf{C}(x_i, y_j) \tag{30}$$

$$+ \varepsilon \sum_{i=1}^{N} \sum_{j=1}^{M} \pi_{i,j} \log \tfrac{\pi_{i,j}}{\alpha_i \beta_j} - \pi_{i,j} + \alpha_i \beta_j$$

$$\text{subject to} \quad \forall i, j, \ \pi_{i,j} \geqslant 0, \quad (\pi \mathbf{1})_i = \sum_{j=1}^{M} \pi_{i,j} = \alpha_i, \quad (\pi^\top \mathbf{1})_j = \sum_{i=1}^{N} \pi_{i,j} = \beta_j$$

$$= \max_{\substack{(f_i) \in \mathbb{R}^N \\ (g_j) \in \mathbb{R}^M}} \sum_{i=1}^{N} \alpha_i f_i + \sum_{j=1}^{M} \beta_j g_j + \varepsilon \sum_{i=1}^{N} \sum_{j=1}^{M} \alpha_i \beta_j \left( 1 - \exp \tfrac{1}{\varepsilon}\left[ f_i + g_j - \mathbf{C}(x_i, y_j) \right] \right) . \tag{31}$$

Up to a small perturbation, the optimal transport problem can thus be reduced to the resolution of (31), a concave maximization problem on the dual vectors $(f_i) \in \mathbb{R}^N$, $(g_j) \in \mathbb{R}^M$ that is smooth and without constraints. The optimal dual potentials encode, implicitly, an optimal transport plan:

$$\pi_{i,j} = \alpha_i \beta_j \, \exp \tfrac{1}{\varepsilon}\left[ f_i + g_j - \mathbf{C}(x_i, y_j) \right] \tag{32}$$

that satisfies the marginal constraints of (30), with an optimal transport cost that reads:

$$\mathrm{OT}_\varepsilon(\alpha_i, x_i, \beta_j, y_j) = \sum_{i=1}^{N} \alpha_i f_i + \sum_{j=1}^{M} \beta_j g_j . \tag{33}$$

**The Sinkhorn algorithm.** The standard Sinkhorn algorithm is equivalent to an alternate maximization of (31) with respect to the dual vectors $(f_i)$ and $(g_j)$. Starting from null potentials $f_i = 0$ and $g_j = 0$, its updates read:

$$f_i \;\leftarrow\; -\varepsilon \log \sum_{j=1}^{M} \beta_j \, \exp \tfrac{1}{\varepsilon}\big[g_j - \mathbf{C}(x_i, y_j)\big] \,, \qquad \forall i \in [\![1, \mathrm{N}]\!] \,, \tag{34}$$

$$g_j \;\leftarrow\; -\varepsilon \log \sum_{i=1}^{N} \alpha_i \, \exp \tfrac{1}{\varepsilon}\big[f_i - \mathbf{C}(x_i, y_j)\big] \,, \qquad \forall j \in [\![1, \mathrm{M}]\!] \,. \tag{35}$$

This method has been (re-)discovered in many applied fields since the 1960's [27, 29, 34, 41, 44, 94, 98, 112, 117], with minor variations on the exact formulation of the regularized problem. Most authors work with the exponentiated variables:

$$u_i \;=\; \exp(f_i/\varepsilon) \qquad \text{and} \qquad v_j \;=\; \exp(g_j/\varepsilon) \,. \tag{36}$$

The dual variables $u = (u_i) \in \mathbb{R}_{>0}^{\mathrm{N}}$ and $v = (v_j) \in \mathbb{R}_{>0}^{\mathrm{M}}$ are then initialized as uniform vectors of 1, with updates that read:

$$u \;\leftarrow\; \frac{1}{K(\beta v)} \qquad \text{and} \qquad v \;\leftarrow\; \frac{1}{K^\top(\alpha u)} \,. \tag{37}$$

In the equations above, the inversions and multiplications $\beta v$, $\alpha u$ are applied coordinate-wise. The $(\mathrm{N}, \mathrm{M})$ matrix $K = (K_{i,j})$ is the Gibbs kernel associated to $\mathbf{C}(x_i, y_j)$ at temperature $\varepsilon$ with coefficients:

$$K_{i,j} \;=\; \exp\big(-\mathbf{C}(x_i, y_j)/\varepsilon\big) \,. \tag{38}$$

When $\mathbf{C}(x_i, y_j) = \tfrac{1}{2}\|x_i - y_j\|^2$, $K$ is a Gaussian kernel matrix of deviation $\sigma = \sqrt{\varepsilon}$: this quantity is best understood as the **blur scale** of the Gaussian smoothing that we apply on the transport plan $\pi_{i,j}$ to lower the complexity of the optimization problem.

**Stabilization.** As detailed in the `sinkhorn_loop_simple` routine below, our library can be used to implement efficiently the exponentiated Sinkhorn updates of (37). In practice though, these iterations may induce numerical overflows and are notoriously unstable when $\sqrt{\varepsilon}$ is too small. Following [23, 37, 70], we rely instead on symmetrized updates performed in the logarithmic domain:

$$\widetilde{f}_i \;\leftarrow\; -\varepsilon \log \textstyle\sum_{j=1}^{M} \beta_j \, \exp \tfrac{1}{\varepsilon}\big[g_j - \mathbf{C}(x_i, y_j)\big] \,, \qquad \forall i \in [\![1, \mathrm{N}]\!] \,, \tag{39}$$

$$\widetilde{g}_j \;\leftarrow\; -\varepsilon \log \textstyle\sum_{i=1}^{N} \alpha_i \, \exp \tfrac{1}{\varepsilon}\big[f_i - \mathbf{C}(x_i, y_j)\big] \,, \qquad \forall j \in [\![1, \mathrm{M}]\!] \,, \tag{40}$$

$$f_i \;\leftarrow\; \tfrac{1}{2}(f_i + \widetilde{f}_i) \,, \qquad \forall i \in [\![1, \mathrm{N}]\!] \,, \tag{41}$$

$$g_j \;\leftarrow\; \tfrac{1}{2}(g_j + \widetilde{g}_j) \,, \qquad \forall j \in [\![1, \mathrm{M}]\!] \,. \tag{42}$$

This robust algorithm is implemented in the `sinkhorn_loop_stable` routine detailed below. Our CUDA engine performs the `.logsumexp()` reduction using an online version of the Log-Sum-Exp trick – with a running maximum – that guarantees numerical stability with a negligible computational overhead.

**Annealing.** In practice, the Sinkhorn loop converges to a set numerical tolerance in $\mathcal{O}(\max_{i,j} \mathbf{C}(x_i, y_j) / \varepsilon)$ iterations. To accelerate convergence, a common heuristic is to let the temperature $\varepsilon$ decrease following an exponential annealing schedule [71]. If $\Delta$ is an estimation of the diameter $\max_{i,j}\|x_i - y_j\|$, $\varepsilon$ is a target temperature and $n_{\mathrm{its}}$ is a prescribed number of iterations, we use decreasing values of the temperature:

$$\varepsilon_n \;=\; \Delta^2 \, q^n \qquad \text{with} \qquad q \;=\; (\varepsilon/\Delta^2)^{1/n_{\mathrm{its}}} \tag{43}$$

at every iteration $n \in [\![1, n_{\mathrm{its}}]\!]$ of the Sinkhorn loop. For faster convergence, the dual potentials are initialized using a closed-form solution of (31) when $\varepsilon = +\infty$:

$$f_i \;=\; \sum_{j=1}^{M} \beta_j \, \mathbf{C}(x_i, y_j) \qquad \text{and} \qquad g_j \;=\; \sum_{i=1}^{N} \alpha_i \, \mathbf{C}(x_i, y_j) \,. \tag{44}$$

Overall, this method usually lets the Sinkhorn loop converge to a satisfying tolerance in 5 to 20 iterations, even for small values of $\varepsilon$.

```python
def sinkhorn_loop_simple(a, x, b, y, eps, nits):
    """
    weights, points,  weights', points'    ->  f(x),   g(y)
    (B, N), (B, N, D), (B, M), (B, M, D),  -> (B, N), (B, M)
    """
    B, N, D = x.shape  # Batch size, source points, features
    _, M, _ = y.shape  # Batch size, target points, features

    # Dual variables, (B, N) and (B, M):
    a, b = a.view(B, N, 1), b.view(B, M, 1)
    u_x, v_y = torch.ones_like(a), torch.ones_like(b)
    # Encoding as symbolic tensors:
    x_i = LazyTensor(x.view(B, N, 1, D))  # (B, N, 1, D)
    y_j = LazyTensor(y.view(B, 1, M, D))  # (B, 1, M, D)

    # Symbolic cost matrix and Gibbs kernel:
    C_ij = ((x_i - y_j) ** 2).sum(-1) / 2  # (B, N, M)
    K_ij = (- C_ij / eps).exp()            # (B, N, M)

    # Sinkhorn iterations:
    for _ in range(nits):
        u_x = 1 / (K_ij     @ (b * v_y))  # (B, N, M) @ (B, M, 1) = (B, N, 1)
        v_y = 1 / (K_ij.t() @ (a * u_x))  # (B, M, N) @ (B, N, 1) = (B, M, 1)

    f_x, g_y = eps * u_x.log(), eps * v_y.log()
    return f_x.view(B, N), g_y.view(B, M)

def sinkhorn_loop_stable(a, x, b, y, eps, nits):
    """
    weights, points,  weights', points'    ->  f(x),   g(y)
    (B, N), (B, N, D), (B, M), (B, M, D),  -> (B, N), (B, M)
    """
    B, N, D = x.shape  # Batch size, source points, features
    _, M, _ = y.shape  # Batch size, target points, features

    # Dual potentials, (B, N) and (B, M):
    f_x, g_y = torch.zeros_like(a), torch.zeros_like(b)
    # Log of the weights, (B, N) and (B, M):
    a_logs, b_logs = a.log(), b.log()

    # Encoding as symbolic tensors:
    # Points:
    x_i = LazyTensor(x.view(B, N, 1, D))  # (B, N, 1, D)
    y_j = LazyTensor(y.view(B, 1, M, D))  # (B, 1, M, D)
    # Dual potentials:
    f_i = LazyTensor(f_x.view(B, N, 1, 1))  # (B, N, 1, 1)
    g_j = LazyTensor(g_y.view(B, 1, M, 1))  # (B, 1, M, 1)
    # Log-weights:
    log_a_i = LazyTensor(a_logs.view(B, N, 1, 1))  # (B, N, 1, 1)
    log_b_j = LazyTensor(b_logs.view(B, 1, M, 1))  # (B, 1, M, 1)

    # Symbolic cost matrix:
    C_ij = ((x_i - y_j) ** 2).sum(-1) / 2   # (B, N, M, 1)

    # Symmetric Sinkhorn iterations, written in the log-domain:
    for _ in range(nits):
        ft_x = - eps * ((g_j - C_ij) / eps + log_b_j).logsumexp(dim=2).squeeze(-1)
        gt_y = - eps * ((f_i - C_ij) / eps + log_a_i).logsumexp(dim=1).squeeze(-1)
        # Use in-place updates to keep a small memory footprint:
        f_x[:] = (f_x + ft_x) / 2
        g_y[:] = (g_y + gt_y) / 2

    return f_x, g_y
```

**Multiscale solvers.** Going further, a recent line of work puts the emphasis on multiscale implementations of the Sinkhorn loop [10, 37, 96]. Following these papers, we use our library to provide a two-scale solver for the regularized optimal transport problem (31). Its behaviour can be described in four steps:

1. We compute **coarse approximations** of the input measures $\alpha$ and $\beta$. In practice, we use a K-means clustering with $N_c = \sqrt{N}$ (resp. $M_c = \sqrt{M}$) clusters on the input point clouds $(x_i) \in \mathbb{R}^{N \times D}$ (resp. $(y_j) \in \mathbb{R}^{M \times D}$): each cluster is represented by its centroid $\overline{x}_i$ with total weight $\overline{\alpha}_i$ (resp. $\overline{y}_j$ with total weight $\overline{\beta}_j$). This corresponds to a quantization of the discrete measures $\alpha$ and $\beta$, as:

$$\sum_{i=1}^{N_c} \overline{\alpha}_i \delta_{\overline{x}_i} \ \simeq \ \sum_{i=1}^{N} \alpha_i \delta_{x_i} \qquad \text{and} \qquad \sum_{j=1}^{M_c} \overline{\beta}_j \delta_{\overline{y}_j} \ \simeq \ \sum_{j=1}^{M} \beta_j \delta_{y_j} \qquad (45)$$

for the weak-$\star$ topology, as measured e.g. by the Wasserstein-2 distance. We sort the $(N, D)$ and $(M, D)$ arrays $(x_i)$ and $(y_j)$ to ensure that all the clusters are contiguous in memory.

2. We start the annealing descent on the **coarse measures**. We use the stabilized iterations of `sinkhorn_loop_stable` on the symbolic $(N_c, M_c)$ cost matrix $\mathbf{C}(\overline{x}_i, \overline{y}_j)$ and update coarse dual vectors $(\overline{f}_i) \in \mathbb{R}^{N_c}, (\overline{g}_j) \in \mathbb{R}^{M_c}$.

3. When the blur scale $\sqrt{\varepsilon_n}$ goes below the largest diameter of the K-means clusters, we perform a **coarse-to-fine extrapolation step**. We use the optimality equations (34-35) to extrapolate the $\overline{f}_i$'s and $\overline{g}_j$'s, supported by the $\overline{x}_i$'s and $\overline{y}_j$'s, onto new values $(f_i) \in \mathbb{R}^N$ and $(g_j) \in \mathbb{R}^M$ supported by the $x_i$'s and $y_j$'s. As discussed in Section 4, we also compute a block-sparsity mask on the symbolic $(N, M)$ cost matrix $\mathbf{C}(x_i, y_j)$: following [96], it corresponds to pruning out pair-wise interactions between clusters such that:

$$\overline{f}_i + \overline{g}_j \ < \ \mathbf{C}(\overline{x}_i, \overline{y}_j) \ - \ \tau \varepsilon_n \ , \qquad (46)$$

where $\tau$ is a cutoff parameter that we set to 5, since $1 \gg \exp(-5)$.

4. We perform the last iterations of the stabilized Sinkhorn loop on the **full point clouds** $(x_i)$ and $(y_j)$: these updates correspond to the values of $\sqrt{\varepsilon_n}$ that range between the average cluster diameter and the target blur value $\sqrt{\varepsilon}$. We use the block-sparsity mask computed at step 3 to prune out negligible interactions from the full $(N, M)$ cost matrix $\mathbf{C}(x_i, y_j)$: this is a GPU-friendly implementation of the **kernel truncation** trick of [96].

The resulting code is too technical to fit in these supplementary materials: we package and fully document this solver on our website (`www.kernel-operations.io/geomloss`) [37, 40].

**Benchmarks.** As discussed above, our library is well suited to research in optimal transport theory: simple algorithms and advanced solvers can all be implemented with symbolic LazyTensors. To showcase the performances of our implementations, we now benchmark several solvers: a baseline linear solver for the exact transport problem, implemented in C++ on the CPU [13, 42]; a stabilized Sinkhorn loop with $1/\varepsilon$ iterations, implemented using either PyTorch JIT or KeOps; a stabilized Sinkhorn loop with annealing and 10 iterations, implemented using either PyTorch JIT or KeOps; a multiscale solver, implemented with KeOps as discussed above. In practice, the parameters of our solvers ensure that the relative error made on the regularized Wasserstein-2 "distance" $\sqrt{2 \cdot \mathrm{OT}_\varepsilon(\alpha_i, x_i, \beta_j, y_j)}$ is always smaller than 1%. This level of accuracy is satisfying for most practical purposes in shape analysis and machine learning.

To illustrate the two main use cases of optimal transport theory in the field, we tackle two separate problems:

1. A high-precision matching in dimension $D = 3$. The input point clouds $(x_i)$ and $(y_j)$ are sampled from the Stanford dragon and are deformed using random affine transformations. They are then centered and normalized: we use an estimate $\Delta = 2$ of the diameter of the configuration in the annealing descent. The blur scale $\sqrt{\varepsilon}$ is set to $0.01$: we retrieve a precise transport plan $\pi_{i,j}$ with (32) that is essentially accurate up to a $< 1\%$ tolerance.

2. A low-precision matching in dimension $D = 25$. The input point clouds $(x_i)$ and $(y_j)$ are sampled from the Glove-25 dataset and are deformed using random affine transformations. They are then centered and normalized: we use an estimate $\Delta = 2$ of the diameter of the configuration in the annealing descent. The blur scale $\sqrt{\varepsilon}$ is set to 0.3: we retrieve a fuzzy transport plan $\pi_{i,j}$ with (32) that captures large-scale deformations while being relatively robust to statistical noise.

Our results are summarized in Table 6 and discussed at the end of the main paper. In practice KeOps consistently improves the runtimes of optimal transport solvers on the GPU: researchers can now scale up their methods to large datasets without memory overflows. As predicted by the theory, multiscale strategies are most useful for large point clouds in low-dimensional spaces ($N \geqslant 100$k, $D \leqslant 3$), while annealing strategies provide a good deterministic baseline in all the other settings.

Table 6: Scaling up optimal transport to large datasets.

| N | D | $\sqrt{\varepsilon}$ | POT Exact | PyTorch Sinkhorn $1/\varepsilon$ its | $\rightarrow$ | **Ours** Sinkhorn $1/\varepsilon$ its | PyTorch annealing 10 its | $\rightarrow$ | **Ours** annealing 10 its | **Ours** multiscale 10 its |
|---|---|---|---|---|---|---|---|---|---|---|
| 1k | 3 | .01 | 121 ms | 2,000 ms | $\rightarrow$ | 241 ms | 1,960 µs | $\rightarrow$ | **82 µs** | 25.7 ms |
| 10k | 3 | .01 | 12.2 s | 203 s | $\rightarrow$ | 7.65 s | 211 ms | $\rightarrow$ | **8 ms** | 26 ms |
| 100k | 3 | .01 | $\infty$ | mem | $\rightarrow$ | 645 s | mem | $\rightarrow$ | 669 ms | **230 ms** |
| 1M | 3 | .01 | $\infty$ | mem | $\rightarrow$ | $\infty$ | mem | $\rightarrow$ | 62 s | **2.70 s** |
| 1k | 25 | .3 | 143 ms | 2,200 µs | $\rightarrow$ | 375 µs | 1,960 µs | $\rightarrow$ | **360 µs** | 36.5 ms |
| 10k | 25 | .3 | 12.6 s | 227 ms | $\rightarrow$ | 35 ms | 211 ms | $\rightarrow$ | **34 ms** | 101 ms |
| 100k | 25 | .3 | $\infty$ | mem | $\rightarrow$ | 3.48 s | mem | $\rightarrow$ | **3.37 s** | 3.40 s |
| 1M | 25 | .3 | $\infty$ | mem | $\rightarrow$ | 319 s | mem | $\rightarrow$ | 338 s | **294 s** |

# F    Structure of the inner KeOps++ engine

This Section provides an overview of the low-level structure of the KeOps engine: more details and explanations can be found on our website (`www.kernel-operations.io`).

**The compilation stack.**    As described in Section 4, effective KeOps computations are triggered by reductions over one of the "symbolic" axes of a `LazyTensor`, at positions $-2$ or $-3$. Calculations are performed by custom binaries that are generated as required by the engine, and stored on the hard drive for later use. Under the hood, the formula $F$ of Eq. (1) is encoded as a string of characters that is attached to the `LazyTensor` object: this simple descriptor is sent to the C++/CUDA compiler via a preprocessor macro through the `cmake` build engine.

In practie, after the compilation step, two dynamic libraries are generated with extension `.so` on Unix, `.dll` on Windows or `.dylib` on MacOS. The first one contains the C++/CUDA functions that perform the actual computation on the GPU, whereas the second one makes the interface between the C++/CUDA code and Python via the PyBind11 library [61]. We note that this second shared object (the binder) can be changed to fit the requirements of other scripted languages: for instance, our gnu/R interface relies on the Rcpp [33] framework. Each binary has a unique name, created using a standard hash function, that identifies the formula $F$ and several other parameters: the Python version, GPU Id, *etc...* KeOps binaries are ultimately gathered in a cache directory that is listed in the `PYTHON_PATH`: they can be imported from Python using the standard `import` statement.

**Building formulas, automatic differentiation.**    Internally, KeOps encodes formulas as recursively templated C++ classes: every single mathematical operation that make up our formulas is defined as a templated `struct` that takes a sub-formula as an input. The recursion ends when the compiler encounters a class that corresponds to a variable or a constant, whose value is known. Every KeOps `struct` that encodes a mathematical operator comes with two attributes:

1. a *forward* function that implements the actual computation in C++/CUDA. This piece of code will be inlined in the final CUDA kernel.

2. a *backward* function that encodes a symbolic expression for the gradient (i.e. the adjoint of the differential), expressed using KeOps recursive templates.

As an example, the element-wise, vector-valued exponential function is encoded with:

```
template < class F >
struct Exp : UnaryOp< Exp, F > {
  // dimension of the output: Exp(F) has the same dimension as F
  static const int DIM = F::DIM

  // Forward: actual computation, to be inlined inside the Cuda code
  static DEVICE INLINE void Operation(TYPE *out, TYPE *in) {
    #pragma unroll
    for (int k=0; k<DIM; k++) { out[k] = exp(in[k]); }
  }

  // Backward: templated expression for the adjoint of the differential
  // operator of Exp w.r.t. the variable V, and applied to GRADIN input
  // vector: ∇_V(Exp(F)).GRADIN = ∇_V(F).(Exp(F)×GRADIN)
  template < class V, class GRADIN >
  using DiffT = typename F::template DiffT< V, Mult< Exp< F >, GRADIN > >;
};
```

Using similar definitions for other mathematical operations, we can then express a Gaussian matrix-vector product:

$$F(x, y, b) = \sum_j \exp(-\|x_i - y_j\|^2)\, b_j \tag{47}$$

as a sum reduction over the index "$j$" of the formula:

```
auto F = Scal< Exp< Minus< SqDist< X, Y> > >, B >
auto SF = Sum_Reduction< F, 0 >  // the 0 flag specifies a reduction over j
```

In the code above, X, Y and B are special classes that represent data loaders (i.e. the variables that are fed to the symbolic formula). `LazyTensor` objects build such templated formulas on their own, whenever required: end-users only have to deal with our high-level `Python` syntax.

The templated structure of our inner engine has two main advantages. First, the code for evaluating the full formula $F$ is built up at compile time, which allows the `C++/CUDA`compiler to optimize the resulting code. Many checks can be performed during the compilation (*e.g.* with `static_assert` expressions) to avoid overheads at run time. Moreover, all loops whose indices are known at compile time (e.g. to compute the norm of a vector of size D) are unrolled aggressively.

Second, we can use the recursive template mechanics to implement a fully-fledged automatic differentiation engine. We recall here that a given KeOps shared object can only compute a single formula. Consequently, in order to compute the gradient $\nabla F$ of a formula $F$, we need to build new shared objects to take care of partial derivatives with respect to all the input variables. As in the forward evaluation, this is done on-the-fly during the call to the Pytorch `.backward()` or `.grad()` methods in a way that is transparent to end-users. For instance, the partial derivative of a Gaussian matrix-vector product with respect to a variable $x$, can be computed by adding the symbolic `Grad< >` operator in front of the formula SF that encodes the sum reduction of a formula F:

```
auto GSF = Grad< SF, X, E >
```

Here, the input variable E is the gradient to back-propagate from the outputs. The effect of the `Grad< >` operator is then surprisingly simple: instead of injecting the code of the forward function, the compiler inlines the code contained in the `DiffT` method – the backward function. The resulting `C++/CUDA` function then outputs the chain rule derivative of SF without hassle.