[Reviews · NeurIPS 2020]

Review 1

Summary and Contributions: The paper presents a python language framework to describe calculations between a pair of point sets that look like distance calculations. Such calculations are common in kernel distance computations, clustering, nearest neighbor search and other methods. The framework tries to convert such code in to high performance code for GPUs. While many of the kernels mentioned in the paper are already implemented well in existing frameworks, by allowing the user to write their own custom calculation, and automatically generating optimized code, the framework gives new tools for experimental research. The evaluation consists of performance comparisons with existing packages on a variety of algorithms that you would expect to benefit from this.

Strengths: The problem is relevant. I have often been in a place where I have tried a new method at small scale, and then need to write fast memory-optimized code to try the method on a larger dataset. This framework nicely folds this into a python framework and automates grunt work. Further, this can be integrated into training pipeline. Done well, I can see this being useful for researchers who don't know how to or do not want to write custom code but want to measure new methods.

Weaknesses: I am not clear on two points: 1. Expressiveness of the framework and scope of optimization: The paper states that it supports computations of the form <aggregate> F(i,j,x_i, y_j), and optimizes for specific patterns of looping over i,j (e.g. block sparse) but is vague regarding the cases that are optimized and those that are not. Can you discuss to what extent you optimize for the categories you listed in related work? 2. Evaluation. Specifically, I am skeptical about Table 2. More details in the "correctness section"

Correctness: I have multiple concerns regarding Table 2: 1. Is your approach bruteforce? Then are you not biasing the comparison by picking tiny datasets? If the benchmark used 10 to 100 million points, as is probably typical in medium scale use cases, brute force will be inefficient. 2. In what sense is FAISS state-of-the-art? At what recall are you measuring FAISS? Why about graph based algorithms? 3. Why has FAISS run out of memory at 10M points? Hasn't it been demonstrated to work on really large datasets. I would request a complete rethinking of Table 2 if this paper were to go ahead. This table also raises concerns that the other tables may not be listing the strongest packages and fair comparison. Updates Thanks for clarifying the scope of your experiments. Please do update the paper as you mention. Some more suggestions, (a) FAISS is popular, but not necessarily SoTA, (b) IVF-PQ and variants are the most compelling algorithms in FAISS (neither GPU-bruteforce nor FAISS-HNSW), (c) it doesn't make sense to compare to a poor baseline at 10^6 points; compare with scalable algorithms if you must, rather than a strawman.

Clarity: Overall, the paper is well written with a good story to tell. The supplement lists all the relevant code snippets for the algorithms in the framework and experimental details.

Relation to Prior Work: Yes, the relation to prior work is stated and the paper's novelty is in automating the distance calculation framework with interesting use cases and automatic code generation.

Reproducibility: Yes

Additional Feedback: 1. I did not understand the compilation aspects of the framework. There is high level mention in the main paper, but no details are to be found in supplement. It is difficult to appreciate the novelty of this aspect without more details. 2. On CPUs, why not use appropriate blocking, rather than the simple reduction scheme. 3. Is it possible to express calculations that are well supported in the current high-performance libraries in your framework and measure the performance gap between your code generation and these high performance libraries? Those set a higher bar than less commonly used algorithmic libraries. 4. How do you assess the usability of this package? This seems like the key contribution of this work. Update Please do add more details about the internals of the framework and compilation to the updated version. This could be insightful for readers.


Review 2

Summary and Contributions: The paper aims at introducing an extension to CNN computing frameworks meant to improve the performance while being transparent to the user. Such extension aspires at operating at the software level, hence not requiring additional ad-hoc hardware to operate.

Strengths: - The proposed solution relies on the CUDA registers to minimize memory transfers (that are responsible for lowering the performance) - The solution is meant to be transparent to the programmers - The application of what described to solutions in Section 5.1-5.3 are interesting.

Weaknesses: - The solution is only useful when a distance between data samples is computed - The Implementation is very weakly described and seems to be more related to a general description of ideas and programming principles. A detailed description of how solution is implement would have been highly appreciated. - The solution foresees that a specific binary is compiled for every new formula-reduction pair. How can this be implemented? Does this significantly affect the performance? - The improvements brought by the proposed solution are visibile only in a limited range of in the number of samples and dimension (hence not being general)

Correctness: The proposed solution is correct in its implementation. Unfortunately, advantages are relevant only for a range of the configuration parameters

Clarity: The paper is well written

Relation to Prior Work: I would encourage authors to more clearly comment why modern toolboxes are not meant for geometric problem (as commented in Section 2-line 71 and in Section 3-line 92). This is a crucial point to be addressed.

Reproducibility: Yes

Additional Feedback: The overall opinion about the paper is that the idea is interesting but the applicability of what proposed is limited. Hence, moving to the mass-use of the proposed extension is not trivial and not easy.


Review 3

Summary and Contributions: The paper presents an ML toolbox for efficient expression and computations with symbolic tensors. The framework is detailed and extensively tested on several datasets.

Strengths: The framework is obviously very useful and will likely have a big impact for learning applications besides the standard setup with discretized kernels. The paper is well written and the method is comprehensively detailed. The authors argue convincingly for the benefits of the framework. I could definitely see the framework be useful in my own research.

Weaknesses: None

Correctness: Yes

Clarity: The exposition is detailed and clear.

Relation to Prior Work: Sufficiently discussed

Reproducibility: Yes

Additional Feedback: In the category of papers on software and implementation, I view this paper very positively.


Review 4

Summary and Contributions: The authors introduce a library for efficient kernel methods based around online computation of entries in interaction matrix and streaming reduction of these entries. It is based around a domain-specific language for kernel expressions, a specialized CUDA code generator, and bindings and integrations for PyTorch, NumPy, R, and MATLAB. The primary advantage of this package over directly using libraries like PyTorch is that it avoids materializing extremely large matrices for pairwise interactions between large numbers of "point clouds" or similar collections of vectors. Instead, it introduces "symbolic matrices", which have the interface of matrices but have entries computed on the fly. The paper makes a comprehensive and detailed performance comparison, including both individual operations like kNN and end-to-end geometric machine learning models like PointNet, demonstrating faster computation and much lower memory usage than systems that use dense matrices.

Strengths: The library itself, the benchmarks presented, and the seamlessness of the integration with PyTorch demonstrated in the code in the supplemental material all point to a substantial amount of engineering work and attention to detail. The authors have identified an important niche in machine learning computation, where conventional array libraries are inadequate, and they've built a solid replacement. Even if the core capability of the library can be summed up in a single formula (Equation 1), the authors convincingly demonstrate the relevance of this pattern to a wide variety of machine learning applications, from dimension reduction to optimal transport. I expect the software package will see use in several NeurIPS submissions in the future, which seems to me as good as any criterion for the relevance of toolkit papers.

Weaknesses: While the approach introduced is impressively broad in applicability, I don't think the paper spends enough time discussing either its limitations or the limitations in other ML compilers that make them a poor fit for kernel problems.

Correctness: The empirical comparisons with other libraries are, in my opinion, quite rigorous, and the overall claim of asymptotically better memory usage and large-constant-factor better throughput seem strong. I would take issue with one aspect. The authors claim that they "implement the exact same algorithms" in the different frameworks, but the foundation of the approach introduced in the paper is an algorithmic distinction. Depending on the generality of the tool and/or its compiler, it may be possible to express the streaming map+reduce used in the ***** toolkit natively in another framework (for instance, using a compiled JAX/XLA loop over points). That wouldn't necessarily be a better or fairer comparison, but it would provide additional information.

Clarity: Yes--the paper provides an impressive amount of motivation, empirical comparison, analysis, and applications both concisely and readably.

Relation to Prior Work: The authors both discuss the implementation differences with and compare the performance of their library to strong baselines in many different application areas. Their results are impressive, especially given that some of the baselines are heavily optimized for specific problems (e.g. FAISS for kNN). I'm wondering if PyTorch-Geometric's main competitor DGL should be an additional comparison point for the geometric deep learning benchmarks; I think it's often faster in practice although it may be too specialized for these architectures. I would like to see more discussion of the similarities and differences between your implementation and deep learning compilers like XLA and TVM. For instance, does your package do just-in-time CUDA code generation/compilation or perform operator fusion? You point out that one advantage is the ability to produce code agnostic to the number of points (i.e., dynamically shaped in some respects); other than this limitation, are deep learning compilers able to generate equivalent programs (i.e., for streaming point cloud operations) or are they missing crucial features or optimizations? (Imagine that the XLA authors want to learn from your paper.)

Reproducibility: Yes

Additional Feedback:

[Author Response · NeurIPS 2020]

We thank the reviewers for their detailed reports and suggestions. We now provide answers to their main questions.

**"usability" (R1)** As showcased in our examples, users only have to cast their tensors as symbolic matrices: this is a
one-line operation, just like with sparse matrices. We stress that our library runs out-of-the-box on *e.g.* a fresh `Google`
`Colab` session: a simple call to "`pip install *******`" is all it takes to get started.

**"only useful for distance computation", "limited applicability" (R2)** We respectfully disagree. While fast distance
computation is core in many geometric ML methods, we show multiple examples that go beyond this first problem.
Our library has gained a sizeable and diverse userbase in fields ranging from theoretical ML to medical imaging and
quantum chemistry. It has been downloaded over 25k times and is now mature to become a standard toolbox for ML.

**"improvements only in a limited range" (R2)** Our library targets computations on data samples that are made up of
$10^3$ to $10^6$ points in dimension 1 to 100. This is relevant to problems such as geometric deep learning, 3D vision, shape
analysis and optimal transport theory. In data sciences, we provide a sizeable performance boost to methods that are
ubiquitous in applied ML and are often used through the `Scikit-learn` library. As pointed out by R1, R3 and R4, our
efficient support for these problems is bound to have a stimulating impact on ML research. Going further, we would
love to provide optimal performance in all settings – from low-resolution 3D shapes to Google-scale datasets – but note
that the rigid structure of `CUDA` registers makes it a very challenging problem for generic symbolic computations.

**"performance gap" (R1), "similarities and differences with deep learning compilers" (R4)** Let us benchmark
a matrix-vector product with an N × N Gaussian kernel matrix $k(x_i, x_j) = \exp(-\|x_i - x_j\|^2)$ for a point cloud
$x_1, \ldots, x_N$ in $\mathbb{R}^3$. `Halide` and `TVM` implement the same streaming computations as our library, whereas `PyTorch` and
`TF-XLA` attempt to optimize a tensorized code – as detailed in Table 5 of the supplementary materials, Python tiling
with `PyTorch` or `XLA` is also inefficient. Our library is extremely competitive for kernel- and distance-related operations
(timings performed on a `Google Colab` session with a K80 GPU, checked with `nvidia-smi`).

|                          | PyTorch   | PyTorch-TPU | TF-XLA    | Halide  | TVM          | **Ours**      |
|--------------------------|-----------|-------------|-----------|---------|--------------|---------------|
| N = 10k                  | 34 ms     | 10 ms       | 23 ms     | 5 ms    | 6 ms         | **2 ms**      |
| N = 100k                 | mem       | mem         | 1,062 ms  | 360 ms  | 282 ms       | **107 ms**    |
| N = 1M                   | mem       | mem         | mem       | 41.3 s  | 26.5 s       | **10.3 s**    |
| Lines of code / interface | 5 / arrays | 5 / arrays | 5 / arrays | 15 / C++ | 17 / low-level | **5 / arrays** |

**"why modern toolboxes are not meant for geometric problems?" (R2)** The example above provides an insight into
this question. Roughly speaking, frameworks like `TVM` or `Halide` demand technical skills to be used efficiently, whereas
`PyTorch` or `TF-XLA` do not prioritize the geometric computations of Section 2: these require significant investments on
specific `CUDA` schemes to avoid memory issues and accelerate computations when the sample size $N > 10^4$.

**"comparison to DGL" (R4)** will be included in the final version.

**"blocking on CPU" (R1)** On CPU, our library implements the simple reduction scheme of Figure 2.a, with a direct
parallelization via OpenMP. We are are working on explicit SIMD support for future releases. For approximate
computations, our block-sparse scheme is available on both CPU and GPU backends.

**"concerns regarding Table 2" (R1)** All our benchmarks are done fairly. We chose FAISS since it is well established,
performs well in the reference "ANN-Benchmarks" (some competitors such as ScaNN have been added after the
NeurIPS deadline) and is one of the only NN-search libraries with an optimized GPU implementation. It comes with
two main backends: `FAISS-GPU`, a bruteforce GPU search similar to ours (useful when $N \leqslant 10^6$) and `FAISS-HNSW`, a
CPU implementation of the graph-based `HNSW` algorithm, with a significant pre-processing time (useful when $N > 10^6$).
We never intended to compete with approximate, graph-based methods for NN-search on very large datasets and made
this clear in our paper (see Sections 2, 4, 5.2 and the legend of Table 2, where we also state that `FAISS-HNSW` is run
with a recall at 90%). We will stress this more in the final version.

**"why FAISS runs out of memory at 10M points?" (R1)** The backend that fails at 10M points is `FAISS-GPU` which
is a bruteforce implementation, not an ANN method. We do not know exactly why it fails.

**"compilation aspects" (R1), "implementation weakly described" (R2), "specific binary is compiled" (R2).** Our
engine handles all reductions of symbolic matrices through the same parallel schemes: `C++` code is generated for each
formula $F(x_i, y_j)$ and is inserted in a templated `CUDA` kernel that implements the reduction scheme of Figure 2.b. As
detailed in Section 4, compilation overheads are not a bottleneck in practice: binaries are stored on the hard drive
for later use. Our engine is implemented in `C++`, which is key to performance: array-centric libraries prevent us
from managing `CUDA` registers. Block-sparse reductions are also efficient: runtimes are proportional to the sparsity
of the block-wise reduction mask, provided that the dimensions of the tiles exceed the `CUDA` block size ($\sim 100$).
Implementation is fully described on our website and will be included in the final version / supplementary materials.

[Meta-Review · NeurIPS 2020]

Reviewers agree that the central problem addressed by this paper is of value, and all reviewers recommend to accept the paper. The rebuttal makes clear that the potential application is beyond just distance calculations, and I agree with the positive reviewers that there is a potential to influence practice in areas of ML across the NeurIPS audience. I disagree that "we never intended to compete with [approx NN]" was "made this clear in the paper (Section 2, 4, 5.2, and the legend of Table 2)". I believe it was rather quite obliquely implied. There is no mention in "limitations" of the consequences of what R1 refers to as the "bruteforce" comparison. I believe that being clearer on this point would not detract at all from the paper, and indeed would strengthen it.